# TopInG: Topologically Interpretable Graph Learning via Persistent Rationale Filtration

Cheng Xin [* 1]  Fan Xu [* 2]  Xin Ding [2]  Jie Gao [1]  Jiaxin Ding [2]

## Abstract

Graph Neural Networks (GNNs) have shown remarkable success across various scientific fields, yet their adoption in critical decision-making is often hindered by a lack of interpretability. Recently, intrinsically interpretable GNNs have been studied to provide insights into model predictions by identifying rationale substructures in graphs. However, existing methods face challenges when the underlying rationale subgraphs are complex and varied. In this work, we propose TOPING: **Top**ologically **In**terpretable **G**raph Learning, a novel topological framework that leverages *persistent homology* to identify persistent rationale subgraphs. TOPING employs a rationale filtration learning approach to model an autoregressive generation process of rationale subgraphs, and introduces a self-adjusted topological constraint, termed *topological discrepancy*, to enforce a persistent topological distinction between rationale subgraphs and irrelevant counterparts. We provide theoretical guarantees that our loss function is uniquely optimized by the ground truth under specific conditions. Extensive experiments demonstrate TOPING's effectiveness in tackling key challenges, such as handling variform rationale subgraphs, balancing predictive performance with interpretability, and mitigating spurious correlations. Results show that our approach improves upon state-of-the-art methods on both predictive accuracy and interpretation quality.

## 1. Introduction

Graph Neural Networks (GNNs) have emerged as a powerful tool for learning graph-structured data in various scientific domains (including chemistry, biology, physics, and materials science), achieving remarkable success in applications of predicting molecular properties (Kamberaj, 2022; Chen et al., 2023), modeling protein-protein interactions (Görmez et al., 2021; Ravichandran et al., 2024; Li et al., 2023), analyzing phase transitions (Qu et al., 2022), characterizing material characteristics (Hu & Latypov, 2024; Sheriff et al., 2024; Gurniak et al., 2024; Xiao et al., 2024), etc. As GNNs are increasingly applied to critical scientific and decision-making tasks, there is a growing need for interpretability and explainability in these models (Zhang et al., 2024a). Scientists and practitioners often ask for not only accurate predictions, but also insights into why and how these predictions are made. This is particularly crucial in scientific applications where understanding the underlying mechanisms and causal relationships is as important as the predictions themselves.

A recent trend in GNN research focuses on enhancing interpretability by developing methods that identify and visualize the nodes, edges, subgraphs, or features most influential or causal for a given prediction. Existing approaches on GNN interpretation can be broadly categorized into two classes (Zhang et al., 2024a): post-hoc explainer methods (Ying et al., 2019; Luo et al., 2020; Schlichtkrull et al., 2021; Wu et al., 2023; Bui et al., 2024) and intrinsically interpretable models (Wu et al., 2022; Miao et al., 2022; Chen et al., 2024). Post-hoc explainer methods analyze a pre-trained GNN model to generate intuitive explanations. These methods enjoy flexibility and can be integrated into different kinds of models. But they might provide explanations that are suboptimal or inconsistent with the model's actual decision-making processes (Miao et al., 2022). On the other hand, intrinsically interpretable models incorporate interpretability directly into the model architecture and training process. A basic intrinsically interpretable GNN model is built upon the graph attention (Veličković et al., 2018) mechanism. But a naïve application of attention weights does not give a reliable interpretation for real graph data (Ying et al., 2019; Yu et al., 2021), as attention weights

---

[*]Equal contribution  [1]Department of Computer Science, Rutgers University, Piscataway, NJ, USA  [2]School of Computer Science, Shanghai Jiao Tong University, Shanghai, China. Correspondence to: Cheng Xin <cx122@rutgers.edu>, Jiaxin Ding <jiaxingding@sjtu.edu.cn>.

*Proceedings of the 42nd International Conference on Machine Learning*, Vancouver, Canada. PMLR 267, 2025. Copyright 2025 by the author(s).

may not always correlate with actual feature importance. Moreover, the trade-off of interpretability and predictive performance (Du et al., 2019) may not be acceptable in real-world applications. To address these challenges, Miao et al. (2022) proposed a stochastic attention mechanism (GSAT) to use the graph information bottleneck (Wu et al., 2020; Tishby et al., 1999) as the target function, employ attention weights to control the information bottleneck, and sample rationale subgraphs using Gumbel-softmax reparameterization. Similarly, Chen et al. (2024) approached interpretation by searching for rationale subgraphs within the framework of subgraph multilinear extension (SubMT) and proposing a graph multilinear net (GMT) for better SubMT approximation. Wu et al. (2022) proposed Discovering Invariant Rationales (DIR), applying interventions on training distributions to obtain invariant causal rationales while filtering out spurious correlations.

Despite these advancements, existing intrinsic methods often assume either explicitly or implicitly that the subgraph rationales are nearly invariant across different instances within the same category of graphs, even a strong one-to-one correspondence between subgraph rationales and predictions. However, this is overly restrictive and unrealistic in many real-world scenarios, where the graph dataset and the downstream tasks exhibit *variform subgraph rationales*, which can vary significantly in form, size, and topology, even among graphs of the same category. For example, in molecular biology, molecules with the same bioactivity can have different functional groups responsible for that activity (Patani & LaVoie, 1996; Brown, 2012). An aromatic ring, a sulfonamide group, or a heterocyclic compound can each be the key substructure leading to the same pharmacological effect in different molecules. In social networks, the structural reasons for a user to be influential vary significantly. An influential user might have high degree, high betweenness centrality, or serving as crucial bridge nodes connecting different communities. Our observations, supported by experiments on a synthetic dataset (see Figure 3 for the results and Appendix E for the dataset construction), show that existing intrinsically interpretable models struggle with such variability. Models obtained under these assumptions may fail to accurately capture the true causal mechanisms underlying the predictions, resulting in unreliable interpretations and suboptimal generalization performance.

To address the above challenges, we propose *Topologically Interpretable Graph Learning* (TopInG), a novel topological approach to intrinsically interpretable GNNs that leverages techniques from topological data analysis to identify stable and persistent rationale subgraphs, effectively handling the variability in subgraph structures. Our method is inspired by the concept of persistent homology, originating from algebraic topology and recently applied to data analysis and machine learning (Wong & Vong, 2021; Yan et al.,

2021; 2022; Zhao et al., 2020; Immonen et al., 2023; Ye et al., 2023; Swenson et al., 2020). Persistent homology studies the dynamics of topological invariants over various scales through filtrations, allowing us to capture all the changes and persistence of topological features in the data.

Based on this foundation, we introduce a new perspective on the rationale subgraph identification problem. We model the graph attention mechanism as an underlying graph generation process, which ideally constructs the rationale subgraph first, followed by the addition of auxiliary structures. We use persistent homology tools to capture and track the representations and life cycles of topological features during the generating process. To effectively distinguish the rationale subgraph from the complement subgraph, we optimize the parameterized generation procedure to enhance the stability of the rationale subgraph. Specifically, our goal is to amplify the topological differences between the rationale subgraph and the complement subgraph, creating a persistent gap in their topological features throughout the generation process. To achieve this goal, we propose a novel self-adjusting topological constraint, *topological discrepancy*, which measures the statistical difference between two graphs with respect to their topological structures. Topological discrepancy serves as a metric to quantify how well the rationale subgraph is preserved and distinguished from the complement subgraph during the filtration process. We also provide a tractable approximation of topological discrepancy and provide theoretical guarantees that our models are able to achieve ground truth as the unique optimal solution under our loss function.

Our main contributions can be summarized as follows:

- We introduce TopInG, a novel intrinsically interpretable GNN framework that incorporates topological data analysis to identify stable rational subgraphs via persistent rationale filtration learning. We propose a new loss function, *topological discrepancy*, to measure the statistical difference between two graphs with respect to their topological structures.

- We provide a tractable approximation of our topological discrepancy and provide theoretical guarantees that our models are able to achieve ground truth as the unique optimal solution under our loss function. This establishes a solid theoretical foundation for the effectiveness of our approach.

- We empirically demonstrate that TopInG improves existing methods in both prediction and interpretation tasks on multiple benchmark datasets. Additionally, we created a synthetic dataset with variform rationale subgraphs to specifically target challenges faced by previous methods. Our results show that TopInG effectively handles such variability, confirming its ability to address this critical challenge.

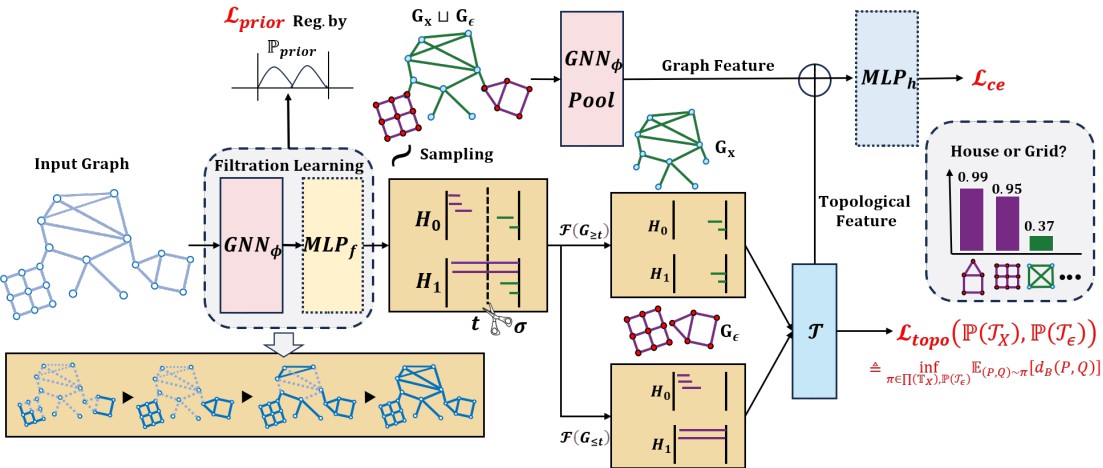

*Figure 1.* An overview of TOPING. A GNN parameterized by $f_\phi$ is used to learn a filtration, from which we sample the subgraphs $G_X$ and $G_\epsilon$. These subgraphs yield topological features through their respective filtrations $\mathcal{F}$. Meanwhile, the combined subgraph $G_X \sqcup G_\epsilon$ is processed by the same GNN (sharing parameters with $f_\phi$) to produce a graph feature. Finally, the topological features $\mathcal{T}$ (which capture global structural information) are combined with the graph feature to form the final graph representation for classification tasks.

The concept of rationalization has also been extensively studied in Natural Language Processing (NLP), where researchers identify text spans as rationales for predictions (Yao et al., 2023; 2024; Gurrapu et al., 2023; Liu et al., 2023; 2024; 2025). However, unlike NLP domains where rationales are typically contiguous text spans, graph domains face the challenge with *variform rationale subgraphs* that vary significantly in size, form, and topology.

## 2. Preliminaries

### 2.1. Graph Neural Networks (GNNs)

Graph neural networks are a class of neural networks designed to operate on graph-structured data. A typical message-passing GNN layer updates node representations by aggregating information from neighboring nodes:

$$h_v^{(l+1)} = \phi(h_v^{(l)}, AGG(h_u^{(l)} : u \in N(v)))  \quad (1)$$

where $h_v^{(l)}$ is the message representation of node $v$ at layer $l$, $N(v)$ is the neighborhood of $v$, AGG is a permutation invariant aggregation function, e.g.: sum, mean, max, and $\phi$ is a non-linear activation function. Some commonly used GNN architectures include Graph Convolutional Networks (GCN) (Kipf & Welling, 2017), Graph Isomorphism Networks (GIN) (Xu et al., 2019), Graph Attention Networks (GAT) (Veličković et al., 2018).

### 2.2. Intrinsically Interpretable Graph Learning

Intrinsically interpretable graph learning aims to build a model simultaneously targeting both prediction and interpretability during the training procedure. Formally, given a collection of labeled graphs $(\mathcal{G}, Y) = \{(G, y_G)\}$, as-

sume each graph $G$ is composed of two edge disjoint subgraphs $G = G_X \sqcup G_\epsilon$ with vertex correspondence for some $G_X \in \mathcal{G}_X$ and $G_\epsilon \in \mathcal{G}_\epsilon$. $\mathcal{G}_X$ and $\mathcal{G}_\epsilon$ are two families of graphs. $\mathcal{G}_X$ is usually a small finite set. Given a graph $G$, $G_X \subseteq G$ is the rationale subgraph that determines the label $y_G$, for some unknown oracle $h^* : \mathcal{G} \to [0, 1]$. $G_\epsilon$ is the noisy or less relevant part of the graph. Both $G_X$ and $G_\epsilon$ are unknown, and they have to be learned from the data. The goal is to predict the label $\hat{y}_G$ for each graph $G$ and simultaneously identify its rationale subgraphs $G_X$.

### 2.3. Topological Data Analysis

Topological Data Analysis (TDA) has emerged as a powerful analytical framework across diverse domains, including machine learning, artificial intelligence, and computational neuroscience. Within graph representation learning specifically, TDA has demonstrated significant capacity to enhance GNNs through the systematic incorporation of topological features (Hofer et al., 2017; 2019; 2020; Dehmamy et al., 2019; Carrière et al., 2020; Horn et al., 2022; Zhao et al., 2020; Carrière & Blumberg, 2020; Zhang et al., 2022; 2024b; Yan et al., 2022; Xin et al., 2023; Mukherjee et al., 2024). A particularly effective methodological tool in this context is *persistent homology*, which provides a rigorous mathematical framework for analyzing the evolution of topological features—such as connected components and cycles—throughout a graph's construction process. This approach enables the quantitative analysis of structural patterns by examining their emergence and persistence across a parameterized filtration of the graph, effectively capturing multi-scale topological information that can be integrated into a learning framework. We give a brief introduction to

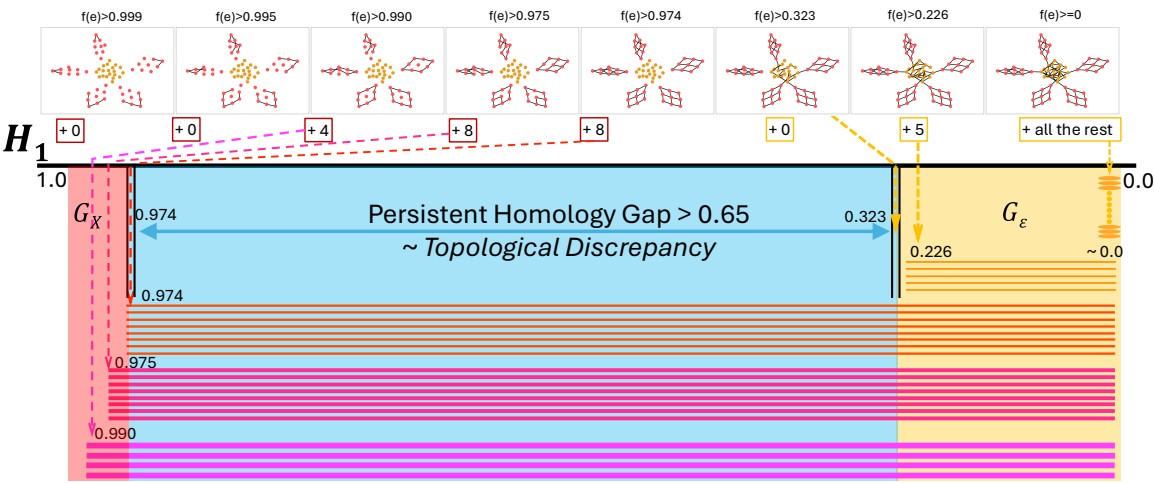

*Figure 2.* The top row is a learned graph filtration on an example graph through our method. Red and yellow points correspond to ground truth rationale subgraph $G_X^*$ and noisy subgraph $G_\epsilon^*$ respectively. Each snapshot is a subgraph of $G$ on edges with filtration values greater than a decreasing threshold value indicated on top of each snapshot. Below each subgraph, the number in small box is the size of cycle basis for the current subgraph, which equals to the dimension of the cycle space, also known as the 1st betti number. The bottom part shows the 1st persistent homology. Each horizontal bar corresponds to a topological feature (basic cycle).

the basic concepts of topological data analysis (TDA) and persistent homology. For a more detailed introduction, we refer readers to (Edelsbrunner & Harer, 2010; Dey & Wang, 2022).

**Graph Filtration**: For an edge-weighted graph $G = (V, E, f : E \to \mathbb{R})$, we can create a sequence of nested subgraphs called a *graph filtration*. For example, assume the edge weight $f$ is normalized, $f(E) \subseteq [0, 1]$, and represents some "importance scores" of edges. We construct a graph filtration $\mathcal{F}(G) := \{G_{\leq t} \mid t \in 1 - f(E)\}$ where $G_{\leq t}$ is the subgraph on edges $e$ with $1 - f(e) \leq t$. Essentially, such graph filtration shows how the graph grows as edges are included in the decreasing order of importance scores.

On a graph filtration, one can track all the connected components and cycles appearing and disappearing (merged with others) during the process. If we encode the lifecycle (birth, death) of each component or cycle as an interval on the real line, it turns out that there is an essentially unique way to represent such information as a multi-set of intervals which is topologically stable and equivalent to a well-studied algebraic structure studied in topological data analysis, known as *persistent homology*. To understand this formally, we first introduce homology vector spaces and then build up to persistent homology.

**Homology Space:** For a given graph $G$, we consider homology vector spaces over the finite field $\mathbb{F}_2 = \{0, 1\}$ (homology spaces in brief). The 0-th homology $H_0(G)$ is defined as the vector space with basis corresponding to all connected components of $G$. The 1-st homology $H_1(G)$ is the vector space defined on the set of all cycles in $G$, with addition operation defined as the symmetric difference of cycles. The

zero vector in $H_1(G)$ is the empty cycle. This vector space is also called the cycle space and its basis is known as cycle basis, which is well studied in graph theory (Horton, 1987; Kavitha et al., 2009; Jungnickel, 2007). For example, for a graph $G = (V, E = \emptyset)$ with no edges, the 0-th homology $H_0(G) = \{0, 1\}^{|V|}$ is a $|V|$-dimensional vector space with basis being the set of all isolated vertices, while $H_1(G) = 0$ is trivial since there are no cycles.

**Persistent Homology:** Starting from a graph filtration $\mathcal{F}(G)$ given by the reversed order of edge weights:

$$\mathcal{F}(G) : \emptyset \subseteq \cdots \subseteq G_{\leq t_1} \subseteq G_{\leq t_2} \subseteq G_{\leq t_3} \subseteq \cdots \subseteq G,$$

if we apply the $p$-homology functor $H_p$ to the graph filtration, each subgraph $G_{\leq t}$ is mapped to a $p$-th homology vector space $H_p(G_{\leq t})$. Each inclusion $G_{\leq t} \subseteq G_{\leq t'}$ naturally induces a linear map $H_p(G_{\leq t}) \to H_p(G_{\leq t'})$. In this way, we get a chain of homology vector spaces $H_p(\mathcal{F}(G))$ connected by linear maps:

$$0 \to \cdots \to H_p(G_{\leq t_1}) \to H_p(G_{\leq t_2}) \to \cdots \to H_p(G).$$

Such structure $H_p(\mathcal{F}(G))$ is called the $p$-th *persistent homology* of the graph filtration $\mathcal{F}(G)$. In this work, we only focus on persistent homologies with $p = 0, 1$, which respectively correspond to the lifecycles of connected components and cycle bases of the graph filtration. This algebraic structure is in fact a graded module over the polynomial ring $\mathbb{F}_2[t]$ (Zomorodian & Carlsson, 2004). By the structure theorem of finitely generated modules over a principal ideal domain, $H_p(\mathcal{F}(G))$ can be uniquely decomposed into a direct sum of cyclic modules. Each indecomposable cyclic module is determined by a pair of numbers $(t_1, t_2)$, which

essentially corresponds to the lifecycle of one persistent topological feature. The multiset of all such pairs is called the *persistence diagram* of the graph filtration, or equivalently, it can be viewed as a multiset of intervals, which is known as the *persistent barcode* of the graph filtration. The persistent barcode (also equivalently the persistence diagram) is a complete discrete invariant of the persistent homology, which means it fully encodes all topological features represented by the persistent homology (Zomorodian & Carlsson, 2004). We refer readers to (Edelsbrunner & Harer, 2010; Zomorodian & Carlsson, 2004; Ghrist, 2008) for further details about persistent homology.

*Remark* 2.1. In this work, one can treat the persistent homology as a topologically stable representation on a sequential input. The representation is differentiable and encodes the evolutions of topological features. See Figure 2 as an example of a graph filtration (cubes on the top line) and its persistent homology (the multiset of intervals shown as horizontal lines in the bottom). We will revisit this example later after our method is introduced.

**Comparing Topological Features:** One can compare two edge-weighted graphs by computing distances between the persistent homologies of their corresponding graph filtrations. A commonly used (pseudo-)metric is the bottleneck distance $d_B$ (Definition B.1 in Appendix B). Recall that persistent homology can be represented as a multiset of points in $\mathbb{R}^2$. The bottleneck distance is essentially a variant of the Wasserstein distance between these two multisets (persistence diagrams) under the $\ell_\infty$-norm. A crucial property of the bottleneck distance is its stability: small perturbations in the input data (e.g., edge weights) lead to small changes in the bottleneck distance, ensuring robustness in downstream applications (see Appendix B for more details). Our proposed topological discrepancy is constructed based on the bottleneck distance and its stability property.

## 3. Method of TOPING

In the following context, for a given $G$, we denote the oracle rationale subgraph and its complement as $G_X^*$ and $G_\epsilon^*$ respectively. We use $G_X$ and $G_\epsilon$ to represent a candidate rationale subgraph and its complement respectively predicted by our model.

In contrast to existing methods, we re-examine the problem from a global perspective through the lens of topology. Our central hypothesis is that: if the prediction task is determined by a core substructure $G_X^*$ belonging to a small family of rationale graphs $\mathcal{G}_X$, then the full graph $G$ can be viewed as being "grown" from this core $G_X^*$ by attaching auxiliary structure $G_\epsilon^*$. The identification of such core rationale substructures is highly non-trivial, as it demands consistency across the generation process and the discovery of common topological patterns across the entire dataset.

To address this, we propose to learn a *rationale filtration*, which represents the importance ordering of edges in a process that simulates an autoregressive generation. By learning to prioritize the edges of the rationale, this approach enables us to identify stable and persistent substructures that are most relevant for predictions. We target a generating process for $G = (V, E)$ that ideally generates the critical rationale subgraph $G_X^*$ first, followed by a less relevant counterpart $G_\epsilon^*$, as the complement.

More precisely, we utilize a backbone GNN as a learnable filtration functional $f_\phi : G \to [0, 1]^{|E|}$ that, for each graph, produces a filtration function $f_\phi^G : E \to [0, 1]$ mapping edges to their importance score. These scores induce an importance ordering on the edges, and consequently, a graph filtration $\mathcal{F}(G) = \{G_0, G_1, \ldots, G_{|E|}\}$, which is an increasing sequence of subgraphs constructed incrementally by adding edges in order of decreasing importance (more important edges introduced earlier in the sequence). By tradition, $G_0 = \emptyset$ and $G_{|E|} = G$. The objective is for the learned filtration functional $f_\phi$ to ideally produce a filtration function on edges whose induced importance ordering is to be consistent with the partitioning into $G_X^*$ and $G_\epsilon^*$, such that $\forall e \in G_X^*$ and $e' \in G_\epsilon^*$, $f(e) > f(e')$.

For notational conciseness, we omit the super- and subscripts for $f = f_\phi^G$ and $\mathcal{F}(G) = \mathcal{F}_\phi(G)$ when the context is clear. We denote $\mathcal{F}(G_{\leq t})$ to be the filtration of the subgraphs on edges with filtration values greater than $1 - t$. Symmetrically, let $\mathcal{F}(G_{\geq t})$ be the filtration of the subgraphs on edges with filtration values smaller than or equal to $1 - t$.

Our approach is founded on ensuring the following property:

**Persistent Homology Gap**: There is a significant difference between the topological features derived from two components of a graph's filtration, $\mathcal{F}(G_{\leq t})$ and $\mathcal{F}(G_{\geq t})$, which are well separated at some threshold value $t \in [0, 1]$. The persistent homologies computed from these respective parts serve as the topological invariants, denoted as $\mathcal{T}_X$ and $\mathcal{T}_\epsilon$.

*Remark* 3.1. The underlying idea of this property is that, when considering the generating process of rationale subgraphs and their irreverent counterparts, their topological structures follow two distinct evolutionary paths. Our methods are designed to statistically capture such a *topological discrepancy*. Figure 2 provides a visual illustration of our method's objective when successfully applied. A well-learned graph filtration (top row) assigns higher importance scores to edges in the ground-truth rationale subgraph $G_X^*$. As a result, the $1^{st}$ persistent homology barcode (bottom row), which tracks the lifecycles of basic cycles, clearly distinguishes the topological features of the learned rationale $G_X$ from those of the complement $G_\epsilon$. The annotation "Persistent Homology Gap" in the figure exemplifies this desired clear separation, where features identified with $G_X$ (long bars appearing earlier that persist significantly) are distinct

from features associated with $G_\epsilon$ (shorter bars introduced later with shorter persistence). Our topological discrepancy measure, introduced subsequently, is designed to quantify and optimize such a distinguishing topological gap.

We denote the induced probability distributions of persistent homologies as $\mathbb{P}(\mathcal{T}_X)$ and $\mathbb{P}(\mathcal{T}_\epsilon)$ respectively.

**Definition 3.2** (Topological Discrepancy). The *topological discrepancy* $d_{\text{topo}}$ between $\mathbb{P}(\mathcal{T}_X)$ and $\mathbb{P}(\mathcal{T}_\epsilon)$ is defined as

$$d_{\text{topo}}(\mathbb{P}(\mathcal{T}_X), \mathbb{P}(\mathcal{T}_\epsilon)) \triangleq \inf_{\pi \in \Pi(\mathbb{P}(\mathcal{T}_X), \mathbb{P}(\mathcal{T}_\epsilon))} \mathbb{E}_{(P,Q)\sim\pi}[d_{\text{B}}(P, Q)]$$

where $\Pi(\mathbb{P}(\mathcal{T}_X), \mathbb{P}(\mathcal{T}_\epsilon))$ is the set of all couplings between $\mathbb{P}(\mathcal{T}_X)$ and $\mathbb{P}(\mathcal{T}_\epsilon)$, and $d_{\text{B}}(P, Q)$ is the bottleneck distance (Definition B.1 in Appendix B) between the persistent homologies $P$ and $Q$.

Essentially, $d_{\text{topo}}$ is the 1-Wasserstein distance between the distributions of persistent homologies $\mathcal{T}_X$ and $\mathcal{T}_\epsilon$ under the metric $d_{\text{B}}$. Now we are ready to introduce the target loss of our model based on our topological discrepancy property.

$$\mathcal{L}(\phi) = \mathbb{E}_G\left[\mathcal{L}_{\text{ce}}(\hat{y}_G, y_G)\right] - \alpha \mathcal{L}_{\text{topo}}(\mathbb{P}(\mathcal{T}_X), \mathbb{P}(\mathcal{T}_\epsilon)) \quad (2)$$

The topological constraint term $\mathcal{L}_{\text{topo}}$ is realized by the topological discrepancy $d_{\text{topo}}$. The prediction loss term $\mathcal{L}_{\text{ce}}$ is the standard cross-entropy loss between the predicted label $\hat{y}_G$ and the ground truth label $y_G$. The predicted label $\hat{y}_G = h_\phi \sigma f_\phi(G)$ is obtained by applying prediction network $h_\phi$ on the subgraph $G_X$ extracted through $\sigma$ from the filtration $f_\phi(G)$. $h_\phi$ and $f_\phi$ share the same backbone $GNN_\phi$ model, which outputs a permutation equivalent representation (node or edge representation). $f_\phi = MLP_f \circ GNN_\phi$ applys a simple multi-layer perceptron (MLP) model to get a 1-dimensional edge representation as the filtration function. $h_\phi = MLP_h \circ Pool \circ GNN_\phi$ first pools the permutation equivalent presentation of $GNN_\phi$ to get a permutation invariant graph representation, then applys another MLP to get the final graph representation for predicting $\hat{y}_G$. Here we omit other details of learnable parameters in the MLP for simplicity. The persistent homologies $\mathcal{T}_X$ and $\mathcal{T}_\epsilon$, as permutation invariant graph representations, are also used in the final representation $MLP_h$ through combining with the graph representation $Pool \circ GNN_\phi$. See Figure 1 as a high-level illustration of the architecture of our model.

### 3.1. Self-adjusted Topological Constraint

In this subsection, we will discuss the construction and properties of our topological features in details. The original construction of $d_{\text{topo}}$ is intractable in general. Here we provide a tractable lower-bound through Kantorovich duality of 1-Wasserstein distance (Villani, 2009) as follows:

**Proposition 3.3.** *Given a set of* 1-*Lipschitz continuous functions,* $\Psi = \{\psi_1, \psi_2, \cdots, \psi_k\}$, *on the space of persistence*

diagrams, $d_{topo}(\mathbb{P}(\mathcal{T}_X), \mathbb{P}(\mathcal{T}_\epsilon))$ *can be lower bounded by:*

$$\max_{\psi \in \Psi} | \mathbb{E}_{P\sim\mathbb{P}(\mathcal{T}_X)}[\psi(P)] - \mathbb{E}_{Q\sim\mathbb{P}(\mathcal{T}_\epsilon)}[\psi(Q)] |$$

**Learnable Vectorization for the Lower Bound** To practically compute the lower bound of $d_{\text{topo}}$, we need a set of 1-Lipschitz continuous functions $\Psi = \{\psi_1, \psi_2, \cdots, \psi_k\}$ that map persistence diagrams to a Euclidean space where expectations are tractable. TDA offers well-studied vectorization methods for this purpose. We adopt the learnable vectorization approach from (Hofer et al., 2019), which represents persistence diagrams as $k$-dimensional vectors. This is achieved by learning $k$ parameterized kernels, termed structure elements, to capture point distributions on the diagrams. These structure elements are designed to be Lipschitz continuous with some constant $C$. Specifically, on a given persistence diagram $p$, we employ the Rational Hat structure element with learnable center $\boldsymbol{c} \in \mathbb{R}^2$ and radius $r \in \mathbb{R}$, which is defined as:

$$\varphi(p; \boldsymbol{c}, r) = \sum_{\boldsymbol{x}\in p} \frac{1}{1 + \|\boldsymbol{x} - \boldsymbol{c}\|_2} - \frac{1}{1 + | |r| - \|\boldsymbol{x} - \boldsymbol{c}\|_2 |}$$

By setting $\psi = \frac{1}{C}\varphi$, we get a 1-Lipschitz continuous representation function as we want. The expectations $\mathbb{E}[\psi(P)]$ are approximated by empirical means over the data in practice. To select the maximum in the lower bound formulation, in our experiments, instead of a simple softmax, we utilize a 2-head attention mechanism to identify and sum the top-2 maxima from the $k$ vectorized representations. We use $k = 8$ Lipschitz continuous representation functions in our experiments. This learnable vectorization, combined with multi-head attention, not only provides an efficient approximation of $d_{\text{topo}}$ but also facilitates a **self-adjusted** focus on data-dependent topological features. This mechanism guides the model to learn the most task-relevant topological information, which we found empirically to enhance training stability and improve performance. All topological representations are Lipschitz continuous and differentiable almost everywhere, enabling end-to-end training. Computations for these representations and their gradients are performed using the codebase from (Zhang et al., 2022).

In the rest of the paper, we use $d_{\text{topo}}$ to denote the lower bound used in practice. Finally, we give the following theorem to show that our loss $\mathcal{L}(\phi)$ with $d_{\text{topo}}$ is guaranteed to be optimized by the ground truth.

**Theorem 3.4.** *Assume* $\forall G$, $|E_X| < |E_\epsilon|$, *and* $G_X^*$ *is minimal with respect to* $y_G$ *in the sense that any subgraph* $G_X \subset G_X^*$ *losses some information of label, then* $\mathcal{L}(\phi)$ *is uniquely optimized by* $f_\phi^*(e) = 1\{e \in G_X^*\}$.

*Remark* 3.5. Note that our guarantee does not depend on any stability or invariance assumptions on $G_X$, therefore, it will not be affected by variform rationale subgraphs in theory. The proof is deferred to the appendix C.

## 3.2. Prior Regularization

Although we present theoretical guarantees, in practice, simply increasing model capacity does not necessarily lead to better performance, and overfitting may still occur. To mitigate this, we introduce a *prior regularization* term on $f_\phi$ that enforces a marginal distribution $\mathbb{P}_{prior}$ over edge filtrations, thereby helping stabilize the training procedure:

$$\mathcal{L}(\phi) \ + \ \beta \, \mathcal{L}_{prior}\big(f_\phi(G), \mathbb{P}_{prior}\big).$$

Concretely, we assume each edge filtration $f_\phi^G \in [0,1]$ follows a *two-mixture Gaussian* distribution:

$$\mathbb{P}_{prior} \ = \ w \, \mathcal{N}(\mu_1, r_1) \ + \ (1-w) \, \mathcal{N}(\mu_2, r_2),$$

where $w, \mu_1, \mu_2, r_1, r_2$ are parameters. Then we can define the prior regularization $\mathcal{L}_{prior}$ via a Kullback–Leibler (KL) divergence term augmented with a penalty to prevent collapsing to a single mode:

$$\begin{aligned}
&\mathcal{L}_{prior}(f_\phi(G), \mathbb{P}_{prior}) \\
=& D_{\mathrm{KL}}[f_\phi(G) \| \mathbb{P}_{prior}] + \gamma(r_1^{-2} + r_2^{-2}) \\
=& - \sum_{e \in G_E} \log(\mathbb{P}_{prior}(f_\phi(G)_e)) + \gamma(r_1^{-2} + r_2^{-2})
\end{aligned}$$

A key insight is that this *two-mixture prior* induces a clustering mechanism on the edge filtrations in $[0,1]$ for $G_x$ and $G_\epsilon$. We note that any choice satisfying $|\mu_1 - \mu_2| > 0$ and a reasonable weighting $w \in (0,1)$ can still maintain a suitable two-cluster separation on $[0,1]$. The exact cluster centers $\mu_1, \mu_2$ matter less than their separation since the edge filtration is learned and used in the topological discrepancy, thanks to the stability property of persistent homology. This approach fundamentally differs from existing methods such as GSAT (Miao et al., 2022) and GMT (Chen et al., 2024), offering greater stability and reduced sensitivity to hyperparameter choices. In practice, we simply fix $w = 0.5$, $\mu_1 = 0.25$, $\mu_2 = 0.75$, and initialize $r_1 = r_2 = 0.25$. We also apply Gumbel-Softmax reparameterization trick (Jang et al., 2017) used in (Miao et al., 2022) to sample subgraphs.

*Remark* 3.6. Although we only talk about edge filtrations, our methods can be applied to filtrations on nodes, edges, or higher-order simplices (faces, tetrahedrons, etc.). In our experiments, we start with the filtration functions on the nodes and then extend the node filtration to the edge filtration by setting $f(u, v) = \min(f(u), f(v))$ or $\max(f(u), f(v))$. This type of filtration is known as upper- or lower-star filtration in TDA. It contains less information in general since node filtrations can only represent $O(|V|)$ much "information" but edge filtrations can represent up to $O(|E|) = O(|V|^2)$ "information". However, it provides more computational efficiency.

## 3.3. Comparing with Related Works

Two works most related to ours are DIR (Wu et al., 2022) and GSAT (Miao et al., 2022). We briefly compared their work with ours. Compared to DIR, our model also considers the distribution of complement graphs of rationale subgraphs, but in a "soft way". Instead of directly storing sample complement subgraphs, our methods can be viewed as storing a distribution of topological summary of complement graphs, which is more efficient. Relative to GSAT, our loss can also be seen as a variational lower bound of the GIB loss. However, we employ a different prior for the rationale subgraph $G_X$ and remove GSAT's hyperparameter $r$. Our topological loss acts as a self-adjusted cut, separating $G_X$ from $G$. In practice, GSAT's attention can collapse to the constant $r$ if not carefully tuned (Chen et al., 2024). By contrast, our prior performs an unsupervised two-Gaussian clustering (akin to $k$-means), preventing such collapse. Empirically, as long as the two means of Gaussian distributions remain distinct, their exact positions have little effect on performance. Thus, we do not need to tune the hyperparamters, and simply fix them at $0.25$ and $0.75$.

# 4. Experiments

We evaluate our proposed method in terms of both interpretability and predictive performance on various benchmark datasets. Our approach, TOPING, demonstrates significant advantages over state-of-the-art post-hoc interpretation methods as well as intrinsic interpretable models across almost all datasets. We will provide a brief introduction to the datasets, baselines, and experiment setups, and leave more details in the Appendix E.

## 4.1. Experimental settings

**Datasets.** We consider eight benchmark datasets commonly used in the graph explainability literature, categorized into three types: *Single Motif*, *Multiple Motif*, and *Real Dataset*. The first two consist of synthetic datasets. *Single Motif* includes BA-2Motifs (Luo et al., 2020), BA-HouseGrid (Amara et al., 2023), SPmotif0.5 and SPmotif0.9 (Wu et al., 2022). These datasets contain graphs with a single type of motif or structural pattern repeated throughout. *Multiple Motif* includes BA-HouseAndGrid, BA-HouseOrGrid (Bui et al., 2024), and BA-HouseOrGrid-nRnd. The last one is a synthetic dataset we create for verifying the variform rationale challenge for existing intrinsic methods (see Appendix E for more details). These datasets involve graphs with multiple types of motifs, thereby increasing the complexity and providing a more challenging scenario for explanation methods. *Real Dataset* include Mutag (Luo et al., 2020) and Benzene (Sanchez-Lengeling et al., 2020). Appendix E.5 visually illustrates sample graphs from each dataset and interpretation results of different models.

*Table 1.* Interpretation Performance (AUC) on test datasets. The shadowed entries are the results when TOPING outperform the means of the best baselines based on the mean-1*std of TOPING.

| | **SINGLEMOTIF** | | | | **MULTIPLEMOTIF** | | **REALDATASET** | |
|---|---|---|---|---|---|---|---|---|
| **METHOD** | **BA-2MOTIFS** | **BA-HOUSEGRID** | **SPMOTIF0.5** | **SPMOTIF0.9** | **BA-HOUSEANDGRID** | **BA-HOUSEORGRID** | **MUTAG** | **BENZENE** |
| GNNEXPLAINER | $67.35 \pm 3.29$ | $50.73 \pm 0.34$ | $62.62 \pm 1.35$ | $58.85 \pm 1.93$ | $53.04 \pm 0.38$ | $53.21 \pm 0.36$ | $61.98 \pm 5.45$ | $48.72 \pm 0.14$ |
| PGEXPLAINER | $84.59 \pm 9.09$ | $50.92 \pm 1.51$ | $69.54 \pm 5.64$ | $72.34 \pm 2.91$ | $10.36 \pm 4.37$ | $3.14 \pm 0.01$ | $60.91 \pm 17.10$ | $4.26 \pm 0.36$ |
| MATCHEXPLAINER | $86.06 \pm 28.37$ | $64.32 \pm 2.32$ | $57.29 \pm 14.35$ | $47.29 \pm 13.39$ | $81.67 \pm 0.48$ | $79.87 \pm 1.61$ | $91.04 \pm 6.59$ | $55.65 \pm 1.16$ |
| MAGE | $79.81 \pm 2.27$ | $82.69 \pm 4.78$ | $76.63 \pm 0.95$ | $74.38 \pm 0.64$ | $99.95 \pm 0.06$ | $99.93 \pm 0.07$ | $99.57 \pm 0.47$ | $96.03 \pm 0.63$ |
| DIR | $82.78 \pm 10.97$ | $65.50 \pm 15.31$ | $78.15 \pm 1.32$ | $49.08 \pm 3.66$ | $64.96 \pm 14.31$ | $59.71 \pm 21.56$ | $64.44 \pm 28.81$ | $54.08 \pm 13.75$ |
| GIN+GSAT | $98.85 \pm 0.47$ | $98.55 \pm 0.59$ | $74.49 \pm 4.46$ | $65.25 \pm 4.42$ | $92.92 \pm 2.03$ | $83.56 \pm 3.57$ | $99.38 \pm 0.25$ | $91.57 \pm 1.48$ |
| GIN+GMT-LIN | $97.72 \pm 0.59$ | $85.68 \pm 2.79$ | $76.26 \pm 5.07$ | $69.08 \pm 5.14$ | $76.12 \pm 7.47$ | $74.36 \pm 5.41$ | **$99.87 \pm 0.09$** | $83.90 \pm 6.07$ |
| GIN+TOPING | $99.57 \pm 0.60$ | $99.24 \pm 0.66$ | $79.50 \pm 3.71$ | **$80.82 \pm 4.22$** | $95.35 \pm 0.95$ | **$88.56 \pm 2.04$** | $95.79 \pm 1.93$ | **$98.22 \pm 0.92$** |
| CINPP+GSAT | $91.12 \pm 4.93$ | $91.04 \pm 6.59$ | $78.20 \pm 4.48$ | $80.24 \pm 3.66$ | $95.17 \pm 2.46$ | $69.30 \pm 2.48$ | $97.27 \pm 0.47$ | $95.40 \pm 3.05$ |
| CINPP+GMT-LIN | $91.03 \pm 5.24$ | $93.68 \pm 4.79$ | $83.23 \pm 4.30$ | $76.40 \pm 2.38$ | $85.08 \pm 3.85$ | $67.91 \pm 5.10$ | **$97.48 \pm 0.81$** | $94.44 \pm 2.49$ |
| CINPP+TOPING | **$100.00 \pm 0.00$** | **$99.87 \pm 0.13$** | **$95.08 \pm 1.84$** | **$92.82 \pm 2.45$** | **$100.00 \pm 0.00$** | **$100.00 \pm 0.00$** | $96.38 \pm 2.56$ | **$100.00 \pm 0.00$** |

*Table 2.* Prediction Performance (Acc.) on test datasets. The shadowed entries are the results when TOPING outperform the means of the best baselines based on the mean-1*std of TOPING.

| | | **REALDATASET** | | **SPURIOUSMOTIF** | | |
|---|---|---|---|---|---|---|
| **MODEL** | **METHOD** | **MUTAG** | **BENZENE** | **b=0.5** | **b=0.7** | **b=0.9** |
| | DIR | $68.72 \pm 2.51$ | $50.67 \pm 0.93$ | $45.49 \pm 3.81$ | $41.13 \pm 2.62$ | $37.61 \pm 2.02$ |
| | GSAT | **$98.28 \pm 0.78$** | **$100.00 \pm 0.00$** | $47.45 \pm 5.87$ | $43.57 \pm 2.43$ | $45.39 \pm 5.02$ |
| GIN | GMT-LIN | $91.20 \pm 2.75$ | **$100.00 \pm 0.00$** | $51.16 \pm 3.51$ | $53.11 \pm 4.12$ | $47.60 \pm 2.06$ |
| | TOPING | $94.20 \pm 5.61$ | **$100.00 \pm 0.00$** | **$52.22 \pm 2.07$** | **$54.46 \pm 5.76$** | **$50.21 \pm 3.22$** |
| | GSAT | **$96.14 \pm 0.67$** | $99.43 \pm 0.54$ | $74.70 \pm 3.37$ | $70.41 \pm 3.44$ | $65.90 \pm 4.18$ |
| CINPP | GMT-LIN | $95.27 \pm 1.36$ | $98.87 \pm 0.92$ | $73.16 \pm 3.51$ | $69.11 \pm 4.12$ | $68.60 \pm 6.06$ |
| | TOPING | $92.92 \pm 7.02$ | **$100.00 \pm 0.00$** | **$79.30 \pm 3.92$** | **$75.46 \pm 4.62$** | **$77.68 \pm 4.64$** |

**Baselines.** We evaluate the interpretability of several methods by differentiating between post-hoc and intrinsic interpretable approaches. The post-hoc methods we compare include GNNExplainer (Ying et al., 2019), PGExplainer (Luo et al., 2020), MatchExplainer (Wu et al., 2023), and Mage (Bui et al., 2024). Additionally, we consider the intrinsic interpretable methods DIR (Wu et al., 2022), GSAT (Miao et al., 2022), and GMT-Lin (Chen et al., 2024), known for their state-of-the-art interpretation capabilities and generalization performance.

**Setup.** Graph Isomorphism Network (GIN) (Xu et al., 2019) is the default backbone GNN used in baseline models. Because our framework, TOPING, is fundamentally topological based, we also implement it on the CINPP (Giusti et al., 2023) backbone to showcase its full capabilities. This will allow our method to naturally extend beyond standard graphs and operate directly on more general filtrations, including those on richer topological domains like simplicial complexes (Bodnar et al., 2021a;b) and hypergraphs.

**Metrics and evaluation.** For interpretation evaluation, we report explanation ROC AUC following (Ying et al., 2019; Luo et al., 2020). For prediction performance, we report classification accuracy for real datasets and SPmotif (Wu et al., 2022) for generalization performance. All the results are averaged over 5 runs tested with different random seeds. All methods adopt the same graph encoder and optimization

protocol to ensure fair comparisons. We employ recommended hyperparameter settings on baseline methods.

### 4.2. Result Comparison and Analysis

**Variform Rationale Challenge.** As shown in Figure 3, the interpretability of two SOTA intrinsic methods decreases drastically when the complexity of rationale subgraphs increases. Our method's performance is much better and more stable on such datasets with variform rationales. See Figure 14 in Appendix E.5 for more visualization results.

**Interpretation performance.** As shown in Table 1, compared to the most post-hoc based methods(in the first row), and latest intrinsic interpretable models(in the second/third row), TOPING has shown significant improvement across almost all datasets. Especially on the Spurious-Motif datasets, which are challenging due to spurious correlations in the training data, we achieve significant improvement over the previous best approach. On the challenging Multiple Motif and Benzene datasets, TOPING even achieves the best performance.

**Prediction performance.** We compare the results of all intrinsic interpretable models training from scratch. Table 2 shows the prediction accuracy on *Real Dataset* and *Spurious Motif*. TOPING significantly outperforms other baseline models on the Spurious-Motif datasets, which exhibit vary-

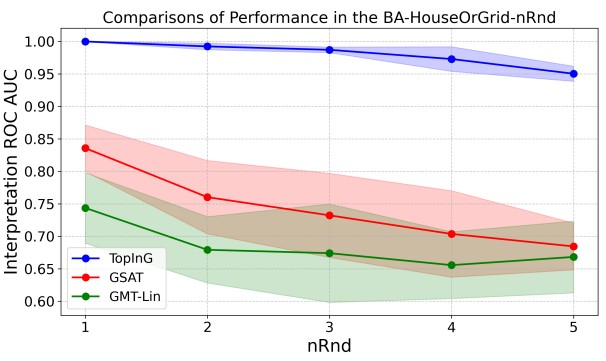

*Figure 3.* In BA-HouseOrGrid-nRnd dataset, as nRnd increases, the complexity of rationale subgraphs increases. Existing SOTA methods struggles on such datasets, while TOPING's performance is much better and stable.

ing degrees of spurious correlations. This supports our claim that the model can more effectively focus on classifying the optimal stable subgraph through persistent rationale filtration learning.

*Table 3.* Ablation studies. We report both interpretation ROC AUC and prediction accuracy.

| METHOD | BA-2MOTIFS | | BA-HOUSEGRID | |
|---|---|---|---|---|
| | ACC | AUC | ACC | AUC |
| TOPING w/o $d_{\text{TOPO}}$ | $100.00 \pm 0.00$ | $97.90 \pm 1.24$ | $89.24 \pm 5.40$ | $92.17 \pm 6.43$ |
| TOPING w/o $\mathcal{L}_{prior}$ | $53.49 \pm 4.03$ | $93.20 \pm 4.61$ | $52.10 \pm 1.72$ | $98.76 \pm 1.53$ |
| TOPING | $100.00 \pm 0.00$ | $100.00 \pm 0.00$ | $100.00 \pm 0.00$ | $99.87 \pm 0.13$ |

**Ablation Studies.** In addition to the interpretability and generalizability analysis, we also conduct further ablation studies to gain a deeper understanding of the results. Table 3 illustrates the usefulness of topological discrepancy and the prior regularizer. Topological discrepancy is essential for identifying stable and complex substructures, and the prior regularizer can be useful in partitioning a graph. We also examine the sensitivity of hyperparameters on the BA-HouseAndGrid dataset. As shown in Fig. 4, TOPING maintains stable performance on different settings of weights of topological discrepancy and prior regularization. The performance decreases on too large or too small weights.

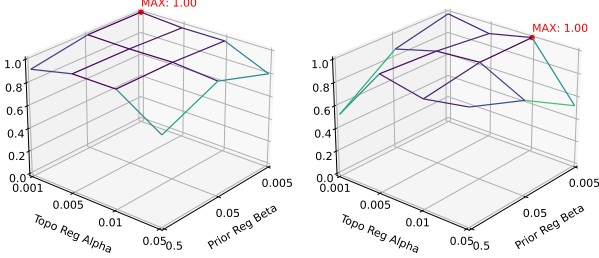

(a) Sensitivity of Interpretation    (b) Sensitivity of Prediction

*Figure 4.* A sensitivity study on BA-HouseAndGrid shows results with the topological constraint coefficient varied from [0.001, 0.005, 0.01, 0.05] and the coefficient of prior regularization term from [0.005, 0.05, 0.5].

## 5. Conclusion

In this work, we introduced TOPING, a novel intrinsically interpretable GNN framework that leverages persistent homology to identify stable rational subgraphs through persistent rationale filtration learning. Our approach introduces a self-adjusted topological constraint, topological discrepancy, to measure the statistical topological difference between graph distributions. We provided theoretical guarantees that our target function can be uniquely optimized by ground truth under certain conditions. Through extensive experiments, we demonstrated that TOPING effectively addresses key challenges in interpretable GNNs, including handling variiform rationale subgraphs, balancing performance and interpretability, and avoiding spurious correlations.

## Acknowledgements

We greatly thank the actionable suggestions given by reviewers. C. Xin and J. Gao acknowledge funding from IIS-22298766, DMS-2220271, DMS-2311064, CRCNS-2207440, CCF-2208663 and CCF-2118953. F. Xu and J. Ding were supported by NSF China under Grant No. T2421002, 62202299, 62020106005, 62061146002.

## Impact Statement

The adoption of powerful but opaque Graph Neural Networks (GNNs) in critical scientific and decision-making domains is hindered by a lack of trustworthy interpretability, as existing methods struggle to explain predictions when the underlying rationales are structurally diverse. This work directly addresses this gap by introducing TOPING, a novel framework that leverages a topological perspective to build more transparent AI systems. By learning to identify stable and persistent rationale subgraphs, TOPING enables researchers to clarify the decision-making process of their models, which is crucial in scientific applications like chemistry and biology where understanding the causal mechanism is as important as the prediction itself. This ability to handle complex and varied rationales leads to more reliable interpretations and improved generalization, providing a new path toward trustworthy AI.

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

# A. List of Notations

In the following, we list the notations for key concepts involved in this paper.

*Table 4.* List of Notations.

| Symbol | Description |
| --- | --- |
| $G = (V, E)$ | A graph with vertex set $V$ and edge set $E$. |
| $G_X$ | Candidate rationale subgraph. |
| $G_\epsilon$ | Candidate noise or less relevant part of the graph. |
| $G_X^*$ | Oracle rationale subgraph. |
| $G_\epsilon^*$ | Oracle noise or less relevant part of the graph. |
| $f_\phi : G \to [0,1]^{|E|}$ | Filtration functional. |
| $\mathcal{F}(G)$ | Graph filtration. |
| $\mathcal{F}(G_{\leq t})$ | Subfiltration consisting of subgraphs with $f(e) \leq t$. |
| $\mathcal{F}(G_{\geq t})$ | Subfiltration consisting of subgraphs with $f(e) \geq t$. |
| $\mathcal{T}$ | Topological invariant (e.g., persistence barcode). |
| $d_{\text{topo}}$ | Topological discrepancy. |
| $d_{\text{bottle}}$ | Bottleneck distance. |
| $d_{\text{wass}}$ | 1-Wasserstein distance. |
| $h_\phi$ | GNN model for prediction. |
| $\sigma$ | Extraction function to separate graph $G$ into $G_X$ and $G_\epsilon$. |
| $\varphi$ | Vectorization function for persistence diagrams. |
| $\mathbb{P}_{\text{prior}}$ | Prior distribution on edge filtration. |
| $\mathcal{L}_{\text{prior}}$ | Prior regularization. |
| $\alpha, \beta, \gamma$ | Hyperparameters for loss function components. |

# B. More details of TDA

**Bottleneck Distance:**

**Definition B.1** (Bottleneck Distance). Let $P_1$ and $P_2$ be two persistent barcodes. A *partial matching* $\pi$ between $P_1$ and $P_2$ is a subset of $P_1 \times P_2$ such that each point in $P_1$ and $P_2$ appears in at most one pair in $\pi$. For any $p = (p_1, p_2) \in \mathbb{R}^2$, denote $\bar{p} = (p_2 - p_1, p_2 - p_1)$. Geometrically, $\bar{p}$ is the closest point of $p$ to the diagonal line $\Delta = \{(x, x) \mid x \in \mathbb{R}\}$. The *bottleneck distance* between $P_1$ and $P_2$ is defined as:

$$d_{\text{B}}(P_1, P_2) = \inf_\pi \max\{ \max_{(p,q) \in \pi} \|p - q\|_\infty, \max_{p \in P_1 \setminus \pi_1} \|p - \bar{p}\|_\infty, \max_{q \in P_2 \setminus \pi_2} \|q - \bar{q}\|_\infty \} \tag{3}$$

where:

- $\pi$ ranges over all partial matchings between $P_1$ and $P_2$

- $\| \cdot \|_\infty$ denotes the $\infty$-norm

- $\pi_1$ and $\pi_2$ denote the projections of $\pi$ onto $P_1$ and $P_2$ respectively

Intuitively, bottleneck distance measures the minimum cost of transforming one barcode to another by moving each point to another point in the other barcode. The cost is measured by the maximum distance between matched points in a partial matching, or the maximum distance between the rest unmatched points to the diagonal line $\Delta$. The bottleneck distance is a metric on the space of persistent barcodes, which is well studied in topological data analysis (Edelsbrunner & Harer,

2010; Zomorodian & Carlsson, 2004; Chazal et al., 2014; Dey & Xin, 2018). It is shown in (Edelsbrunner & Harer, 2010) that the bottleneck distance is a stable metric on the space of persistent barcodes, which means that small perturbations in the input data will not change the bottleneck distance too much. This property is crucial for the stability of the persistent homology, which is a key property for the robustness and differentiability of all vector representations based on persistent homology (Dey & Xin, 2021; 2022; Xin, 2023).

# C. Missing Proofs

**Proof of Theorem 3.4**

*Proof.* By the assumption we know that the first term can only be optimized by $G_X \geq G_X^*$. We just need to show that $d_{\text{topo}}$ is uniquely maximized by $G_X^*$ among those $G_X \geq G_X^*$. In other words, we could assume that we have already restricted $f_\phi$ to the region that satisfies $f_\phi|_{E_X^*} > 0.5 + \delta$ (the partition threshold $t = 0.5$ is fixed).

For a given $G$ and a fixed partition $G_X \sqcup G_\epsilon$ determined by some $f_\phi$, let $p_0, p_1$ be the 0-th and 1-st persistence diagrams, and $q_0, q_1$ be the 0-th and 1-st persistence diagrams. Observe that the bottleneck distance between the 0-th persistence diagrams $d_B(p_0, q_0)$ is maximized when

$$f_\phi(e) = 1\{e \in G_X\}. \tag{4}$$

The reason is that since we only care about edge filtrations, the filtration values on nodes can be viewed as some global minimum constant value which is commonly set to be time 0 (or more precisely, 1 for $G_X$ and 0.5 for $G_\epsilon$ since we build the filtration in the reversed ordering of importance). Then since $|E_\epsilon| > |E_X| \implies |q_0| > |p_0|$, we hope to maximize the death times of points in $q_0$ and minimize the death times of points in $p_0$ to maximize $d_B(p_0, q_0)$, which gives us the constant filtration function $f_\phi(e) = 1\{e \in G_X\}$ on each partition. Then, for constant filtration functions, the induced graph filtrations are essentially reduced to static graphs, and in consequences, persistent homology is essentially reduced to homology. For 0-degree homology, we just need to compare the 0-th Betti numbers $\beta_0^\epsilon$ and $\beta_0^X$ between $G_\epsilon$ and $G_X$. In that case, $d_B(p, q) = C(\beta_0^\epsilon - \beta_0^X) = C(|E_\epsilon| - |E_X|) = C(|G_E| - 2|E_X|)$ for some constant $C$ independent of $\phi$ or $G$. This is maximized when $G_X = G_X^*$.

The rest is to check the bottleneck distance $d_B(p_1, q_1)$ on 1-th persistence diagrams. In a similar way one can check that $d_B(p_1, q_1)$ should be maximized for some constant filtration function. Then the problem is again reduced to compare the 1-degree homology between $G_X$ and $G_\epsilon$. That is $|\beta_1^X - \beta_1^\epsilon|$. However, observe that $|\beta_1^X - \beta_1^\epsilon| \leq \beta_1$ for $\beta_1$ be the 1-st Betti number of the original graph. By the property of the Euler characteristic on a connected graph we know that $\beta_1 \leq |E| - |V| + 1 \leq |E| \leq |V|^2$. Therefore, $d_B(p_1, q_1) \leq M$ for some large enough $M$ over the whole dataset.

Based on that, since $d_{\text{topo}}$ is essentially a weighted sum of $d_B$ on both 0-th and 1-st persistence diagrams, we just need a large enough constant scaling factor on 0-th persistence diagrams. Then it can been guaranteed that our $d_{\text{topo}}$ is optimized by $G_X^*$ with $f_\phi^*(e) = 1\{e \in G_X^*\}$. Such constant factor can be easily learned by our neural networks, or fixed by hand in the model. $\square$

# D. Limitation

One limitation of our model is the computational cost. Currently the bottleneck is limited by the computation of the topological invariants. In theory, the time complexity of the persistent homology computation is $O(n^\omega)$, where $n$ is the number of simplices (nodes for degree 0, edges for degree 1, and faces for degree 2) and $\omega \leq 2.371552$ is the matrix multiplication exponent (Williams et al., 2024). Although on graphs, the 0-th and 1-st persistent homology can be computed much faster in $O(n \log n)$ time, the main bottleneck is not in theoretical computational complexity, but in practical implementation.

**Practical Runtime Analysis.** To provide concrete insights into the computational overhead, we measured the actual training time on representative datasets. On BA-2Motifs, each training epoch takes approximately half a minute, while on SPMotif (a more complex and larger dataset), the runtime is approximately 10 minutes per epoch. All experiments were conducted on a single RTX 4090 GPU. Importantly, our method consistently converges within 20 epochs across all datasets, in contrast to baseline methods that usually require 50-100 epochs to converge.

While our method is relatively slower per epoch due to TDA incorporation, this overhead is justified by significant performance gains and faster convergence. The reduced number of required epochs (20 vs. 50-100 for baselines) partially

compensates for the per-epoch computational cost, making the total training time competitive with baseline methods while achieving consistently superior interpretability performance.

**Implementation Bottlenecks and Future Directions.** The current software package cannot fully utilize the parallel computation power of GPUs. The data transfer between GPU memory and CPU memory takes much I/O cost. Maybe some system-level optimization based on the CUDA framework can help. Some attempts have been made to use GPU to accelerate the computation of persistent homology (Zhang et al., 2020), but the performance is still not satisfactory enough. Another possible solution is to use some approximation algorithms to compute the topological invariants. For example, some efficient sparsification methods (Dey et al., 2019), or pretained NNs for computing persistent homology (Yan et al., 2022). We leave these problems for the future.

# E. More Details about the Experiments

## E.1. Datasets

**Mutag** (Kazius et al., 2005): The dataset involves a task of predicting molecular properties, specifically determining whether a molecule is mutagenic. The functional groups -NO2 and -NH2 are regarded as definitive indicators that contribute to mutagenicity, as noted by (Luo et al., 2020).

**Benzene** (Sanchez-Lengeling et al., 2020): The dataset comprises 12,000 molecular graphs sourced from ZINC15 (Sterling & Irwin, 2015). The objective is to identify the presence of benzene rings within a molecule. The carbon atoms in these benzene rings serve as the ground-truth explanations.

**BA-2Motifs** (Luo et al., 2020): The dataset involves a binary classification task in which each graph combines a Barabasi-Albert base structure with either a house motif or a five-cycle motif. The graph's label and ground-truth explanation are based on the motif it includes.

**SPmotif** (Wu et al., 2022): The dataset consists of graphs that merge a base structure with a motif. Each graph is manually infused with a spurious correlation between the base and the motif. The graph's label and the ground truth explanation are determined by the motif it contains. Specifically, each graph comprises a base graph $\bar{G}_S$ (tree, ladder, or wheel, encoded as 0, 1, 2) and a motif $G_S$ (cycle, house, or crane, also encoded as 0, 1, 2). The label is solely determined by $G_S$, but a spurious correlation is introduced between the label and $\bar{G}_S$. During training, $G_S$ is sampled uniformly, while $\bar{G}_S$ is sampled with:

$$P(\bar{G}_S) = \begin{cases} b, & \text{if } \bar{G}_S = G_S \\ \frac{1-b}{2}, & \text{otherwise} \end{cases}$$

Here, $b \in [\frac{1}{3}, 1]$ controls the degree of spurious correlation; $b = \frac{1}{3}$ implies independence. We consider $b = 0.5, 0.7,$ and $0.9$. For testing, $\bar{G}_S$ and $G_S$ are randomly paired to assess overfitting to spurious correlations.

**BA-HouseGrid**: The house and grid motifs are chosen because they do not have overlapping structures, such as those found in the house and $3 \times 3$ grid.

**BA-HouseAndGrid** (Bui et al., 2024): Each graph is based on a Barabasi-Albert structure and may be linked with either a house motif or a grid motif. Graphs that contain both types of motifs are labeled as 1, while those containing only one type are labeled as 0. Note that each motif appears at most once in each graph.

**BA-HouseOrGrid** (Bui et al., 2024): Similar to BA-HouseAndGrid, graphs that contain either house motif or grid motif are labeled as 1, while those containing neither type are labeled as 0. Note that each motif appears at most once in each graph.

**BA-HouseOrGrid-nRnd**: Similar to BA-HouseOrGrid, graphs that contain either n house motifs or n grid motifs are labeled as 1, where n is a random integer between 1 (inclusive) and n (inclusive). More formally:

- **Label Assignment**:
$$P(\text{Label} = 1) = 0.5, \quad P(\text{Label} = 0) = 0.5$$

- **For Label = 1**: Given $n \in \mathbb{Z}^+$, for each $i \in \{1, 2, \ldots, n\}$, the three possible manifestations are:
$$P(i \times \texttt{grid} + i \times \texttt{house}) = \frac{1}{6n},$$

$$P(i \times \texttt{grid}) = \frac{1}{6n},$$

$$P(i \times \texttt{house}) = \frac{1}{6n}.$$

When `grid` and `house` appear simultaneously, their counts are equal. We do not consider cases where houses and grids appear simultaneously in different quantities. To ensure a balanced dataset and to avoid potential bias in model training and evaluation, we first guarantee that the number of graphs with label 0 and label 1 is equal. Furthermore, within label 1, we generate an equal number of graphs for the above three manifestations. Therefore, the entire dataset maintains a balanced distribution across subcategories.

## E.2. Details on Hyperparamter Tuning

### E.2.1. BACKBONE MODELS

**Backbone Architecture.** We use a two-layer GIN (Xu et al., 2019) with 64 hidden dimensions and 0.3 dropout ratio for all baselines. We use a three-layer CINpp (Giusti et al., 2023) with 64 hidden dimensions and 0.15/0.3 dropout ratio for TOPING. For all datasets, we directly follow (Giusti et al., 2023) using enhanced Topological Message Passing scheme including messages that flow within the lower neighbourhood, the upper neighbourhood and boundary neighbourhood of the underlying cell complex. Considering that the largest chordless cycle for most interpretable motifs is equal to 5 (the BA-2Motifs dataset includes a 5-cycle, while most of the other motifs have chordless cycles with a maximum length of 4), we lift the maximum length of a chordless cycle to 5 as the cell(dim=2).

**Data Splits.** For BA synthetic datasets, we follow the previous work (Miao et al., 2022; Chen et al., 2024; Bui et al., 2024) to split them into three sets(80%/10%/10%). For SPmotifs and real datasets, we use the default splits.

**Evaluation.** We report the performance of the epoch with the highest validation prediction accuracy and use these models as the pre-trained models. If multiple epochs achieve the same top performance, we choose the one with the lowest validation prediction loss.

## E.3. More Comparison Results

To further evaluate the effectiveness of TOPING modules and compare with the previous intrinsic interpretable baselines, we additionally conduct experiments from the perspectives of constraints and regularization on both GIN and CINpp backbones. The results are given in the table 5. Details are as follows.

**Constraint.** Central to existing self-interpretability is the incorporation of the information bottleneck principle into the GNN architecture. We follow previous works to re-implement this infomation constraint under the author-recommended hyperparameters for a fair comparison. The $\lambda$ of information regularizer is set to be 1. As for topological constraint, we set the coefficient to 0.01 to achieve the best performance, which aligning with Figure 4.

**Regularization.** For marginal prior regularization, the choice of information constraint is a KL divergence regularizer. Specifically, for every graph $G \sim \mathbb{P}_G$ and every undirected edge $e$ in $G$, we sample $\alpha_e \sim Bern(r)$ where $r \in [0, 1]$ is a hyperparameter. The formulation of the mutual information regularizer is:

$$D_{KL}\left(\text{Bern}(\alpha_e) \parallel \text{Bern}(r)\right) = \sum_e \left[\alpha_e \log \frac{\alpha_e}{r} + (1 - \alpha_e) \log \frac{1 - \alpha_e}{1 - r}\right] \tag{5}$$

$r$ is initially set to 0.9 and gradually decay to the tuned value 0.7. We adopt a step decay,where r will decay 0.1 for every 10 epochs. As it is mentioned in TOPING, we employ a mixture of two Gaussian distributions to establish a prior regularization term on $G_X$ and $G_\epsilon$, as outlined in Section 3.2. It is noteworthy that our choice aligns with the use of a standard Gaussian distribution as the latent distribution in variational auto-encoders. For fair comparison, we set the coefficient of Gaussian regularizer to 1.

Different components have different effects. Once could select the best combination and train the new architecture to better extract the subgraph information. Moreover, We provide more discussions and analysis about the the results in the table 5. Specifically,

*Table 5.* More Comparison Results

| Backbone | Constraint | Regularization | Prediction | | | | Interpretation | | | |
|---|---|---|---|---|---|---|---|---|---|---|
| | | | **BA-2Motifs** | **SPMotif0.5** | **BA-HouseOrGrid-2Rnd** | **Benzene** | **BA-2Motifs** | **SPMotif0.5** | **BA-HouseOrGrid-2Rnd** | **Benzene** |
| GIN | Info. | Bern | $100.00 \pm 0.00$ | $47.45 \pm 5.87$ | $95.25 \pm 1.60$ | $100.00 \pm 0.00$ | $98.74 \pm 0.55$ | $74.49 \pm 4.46$ | $76.02 \pm 3.64$ | $91.57 \pm 1.48$ |
| | Topo. | Gauss | $100.00 \pm 0.00$ | $50.22 \pm 2.07$ | $91.35 \pm 1.83$ | $100.00 \pm 0.00$ | $99.57 \pm 0.60$ | $79.50 \pm 3.71$ | $88.74 \pm 1.70$ | $98.22 \pm 0.92$ |
| | Topo. | – | $89.35 \pm 5.41$ | $42.80 \pm 5.31$ | $88.41 \pm 1.51$ | $98.35 \pm 0.93$ | $95.79 \pm 3.30$ | $75.92 \pm 2.98$ | $87.88 \pm 2.18$ | $96.54 \pm 0.82$ |
| | – | Gauss | $100.00 \pm 0.00$ | $45.95 \pm 3.02$ | $92.87 \pm 1.88$ | $98.96 \pm 0.30$ | $98.06 \pm 1.81$ | $72.95 \pm 2.45$ | $85.28 \pm 1.98$ | $86.08 \pm 2.68$ |
| CINpp | Info. | Bern | $100.00 \pm 0.00$ | $63.35 \pm 6.06$ | $100.00 \pm 0.00$ | $100.00 \pm 0.00$ | $91.12 \pm 4.93$ | $78.20 \pm 4.48$ | $75.98 \pm 7.09$ | $95.40 \pm 3.05$ |
| | Topo. | Gauss | $100.00 \pm 0.00$ | $79.30 \pm 3.92$ | $100.00 \pm 0.00$ | $100.00 \pm 0.00$ | $100.00 \pm 0.00$ | $95.08 \pm 0.82$ | $100.00 \pm 0.00$ | $100.00 \pm 0.00$ |
| | Topo. | – | $53.49 \pm 4.03$ | $61.18 \pm 3.20$ | $91.83 \pm 6.30$ | $98.52 \pm 1.40$ | $93.20 \pm 4.61$ | $92.10 \pm 3.32$ | $97.78 \pm 1.54$ | $98.96 \pm 1.66$ |
| | – | Gauss | $100.00 \pm 0.00$ | $79.81 \pm 4.39$ | $88.84 \pm 4.93$ | $100.00 \pm 0.00$ | $97.90 \pm 1.24$ | $89.48 \pm 2.54$ | $98.16 \pm 1.25$ | $94.12 \pm 3.49$ |

- When integrating GNN models with enhanced topological message passing scheme, robust explainability methods should be able to adapt accordingly and achieve higher and more stable performance, because stronger expressive power often means that each edge in the graph can be more easily distinguished. However, the performance of graph information bottleneck framework remains highly degenerated in simple task BA-2Motifs and complicated task BA-HouseOrGrid-2Rnd. Due to the formulation 5, there exists a trivial solution where all values of $\alpha_e$ converge directly to the given value of $r$. CINpp employs distinct perceptrons for each layer of the network and each dimension of the complex. It iteratively performs message passing for different types of adjacency. This unique architecture can lead information constraint more easily to zero loss, i.e., $\alpha_e = r$. It will result in the inability to distinguish between interpretable subgraphs and noise subgraphs.

- The topological constraint with a Gaussian prior distribution significantly outperforms both GIN and CINpp across all tasks, although it occasionally slightly reduces accuracy. Equipped with a more powerful expressive model, TOPING can easily learn persistent rationale filtration through topological discrepancy loss. One surprising result is that using only the Gaussian distribution yields highly competitive results in interpreting spurious motif datasets and the BA-HouseOrGrid-nRnd datasets. Nevertheless, the interpretation performance on Benzene dataset remains highly degenerated.

### E.4. Regarding the result of MUTAG

The results of MUTAG can be attributed to the uniquely simple structure of MUTAG's rationale subgraphs. MUTAG's rationale subgraphs consist of just two edges sharing a common node (the functional groups -NO2 and -NH2). This represents the simplest possible non-trivial subgraph structure, lacking the topological complexity present in other datasets. Specifically:

- There are no cycles (1-homology features)

- The 0-homology structure (connectivity) is nearly trivial

- The rationale can be identified primarily through node/edge features rather than topological structure

In such cases, our topological discrepancy measure, which excels at capturing complex structural patterns, may introduce unnecessary complexity by analyzing features (like cycle bases) that aren't relevant to the true rationale. The model ends up relying more heavily on the prediction loss other than the interpretability regularization.

Following up on our previous response regarding MUTAG performance, we conducted additional experiments that provide compelling evidence for our analysis. Our investigation revealed that the initial lower performance on MUTAG stemmed from incorporating both 0th and 1st dimensional persistent homology features. However, the rationale subgraphs of MUTAG —- primarily NO2 and NH2 functional groups —- have relatively simple structures. Therefore, tracking higher-dimensional topological features like cycles introduced unnecessary complexity that hurt the model's performance.

As shown in the Table 6, TopInG-0 achieves the second-best performance in both interpretability (AUC) and prediction (ACC) compared to baseline interpretable GNN models. These results validate our analysis and demonstrate that our approach remains highly competitive when properly configured for molecular datasets with simpler structural patterns.

*Table 6.* Comparison of interpretable GNN on MUTAG dataset.

| Method | AUC | ACC |
| --- | --- | --- |
| DIR | $64.44 \pm 28.81$ | $68.72 \pm 2.51$ |
| GSAT | $99.38 \pm 0.25$ | $98.28 \pm 0.78$ |
| GMT-LIN | $99.87 \pm 0.09$ | $91.20 \pm 2.75$ |
| TopInG | $96.38 \pm 2.56$ | $92.92 \pm 7.02$ |
| TopInG-0 | $99.40 \pm 0.07$ | $95.18 \pm 2.24$ |

### E.5. Interpretation Visualization

We provide visualization of the learned interpretabel subgraphs by GSAT, GMT-LIN and TOPING in the different datasets. The transparency of the edges shown in the figures represents the normalized attention weights learned by interpretable method. Note that we no longer need min-max normalization like (Miao et al., 2022) for better visualization, we can directly use edge attention to visualize through rational filtration learning, because persistent homology gap has guaranteed that our edge attention is easy to be distinguished.

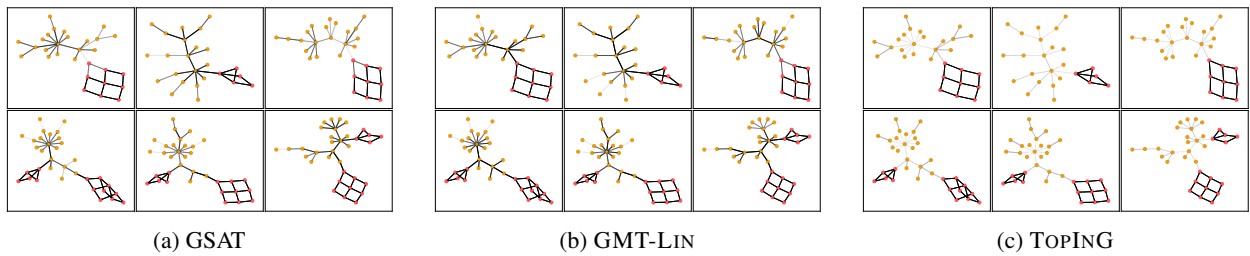

(a) GSAT                   (b) GMT-LIN                   (c) TOPING

*Figure 5.* Learned interpretable subgraphs by GSAT, GMT-LIN and TOPING on BA-HouseAndGrid. Figures in each row belong to the same class. Nodes colored red are ground-truth explanations.

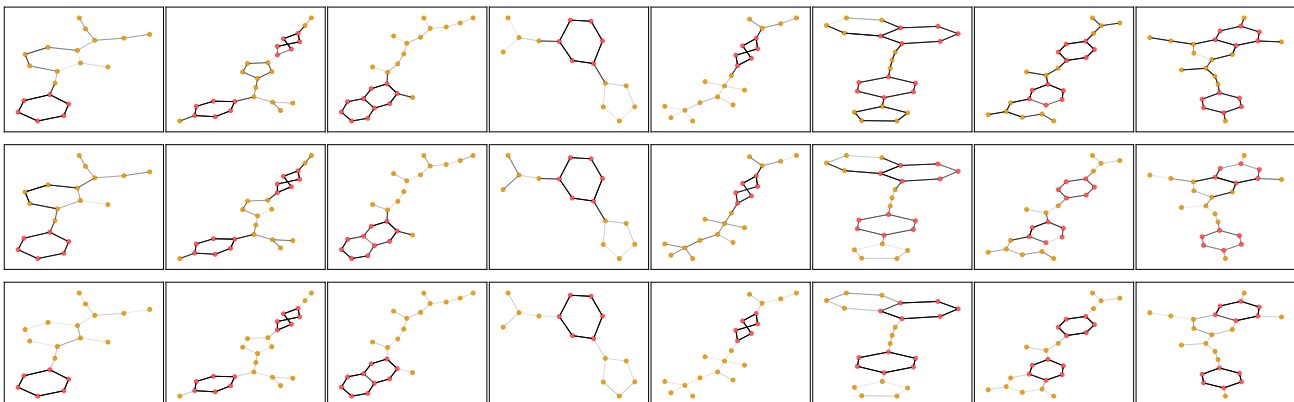

*Figure 6.* Visualizing attention of GSAT(first row), GMT-LIN(second row) and TOPING (third row) on Benzene. Figures in the same column represent an identical graph. Nodes colored red are ground-truth explanations.

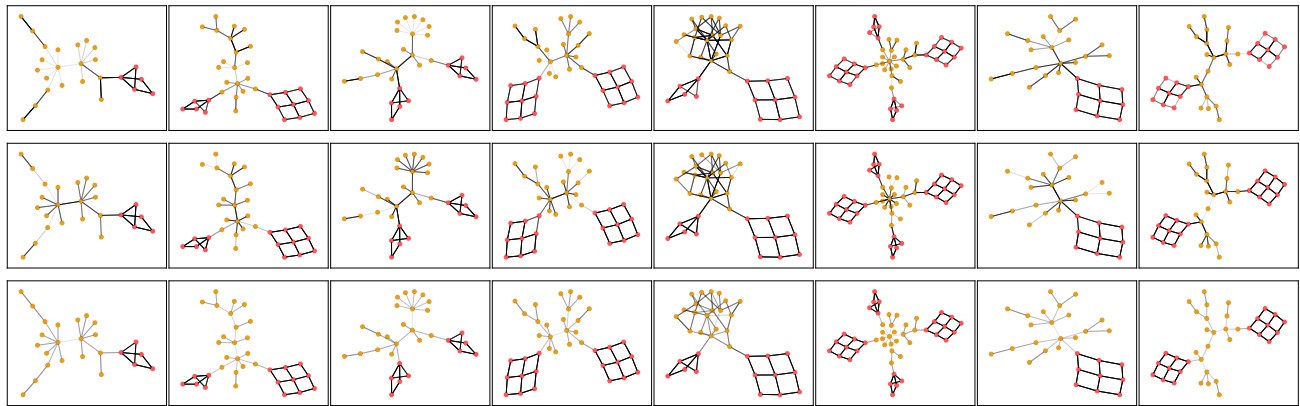

*Figure 7.* Learned interpretable subgraphs by GSAT (first row), GMT-LIN(second row) and TOPING(third row) on BA-HouseOrGrid-2Rnd. Figures in the same column represent an identical graph. Nodes colored red are ground-truth explanations.

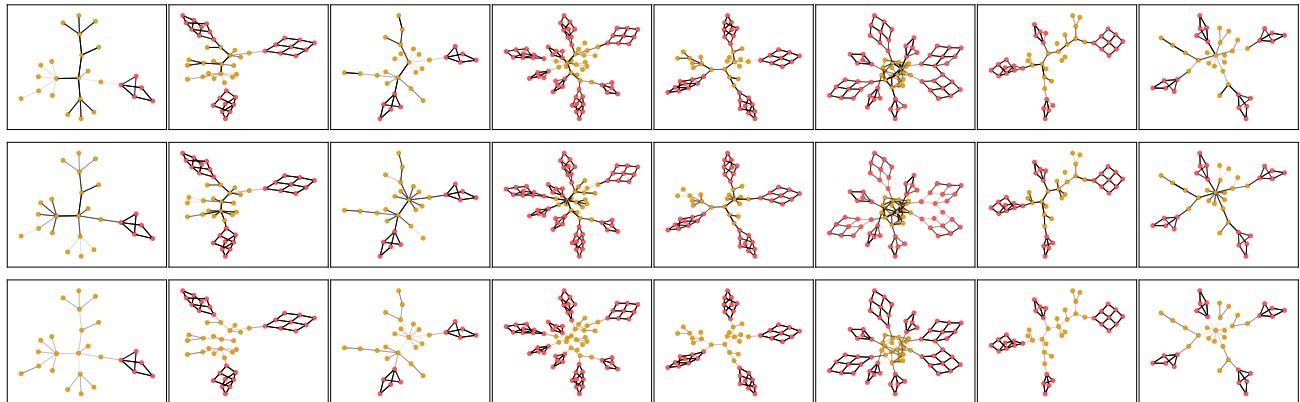

*Figure 8.* Learned interpretable subgraphs by GSAT (first row), GMT-LIN(second row) and TOPING(third row) on BA-HouseOrGrid-4Rnd. Figures in the same column represent an identical graph. Nodes colored red are ground-truth explanations.

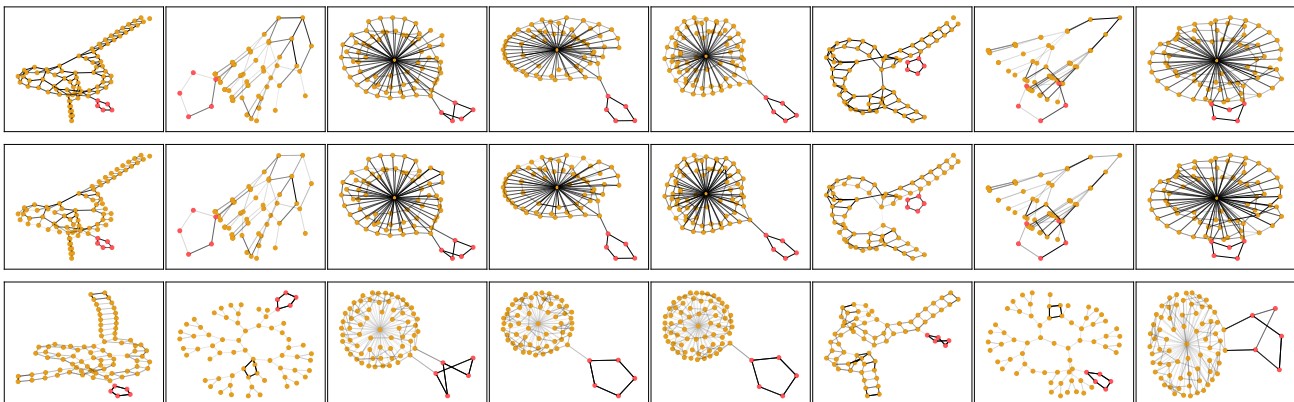

*Figure 9.* Learned interpretable subgraphs by GSAT (first row), GMT-LIN(second row) and TOPING(third row) on SPmotif0.9 class 0. Figures in the same column represent an identical graph. Nodes colored red are ground-truth explanations.

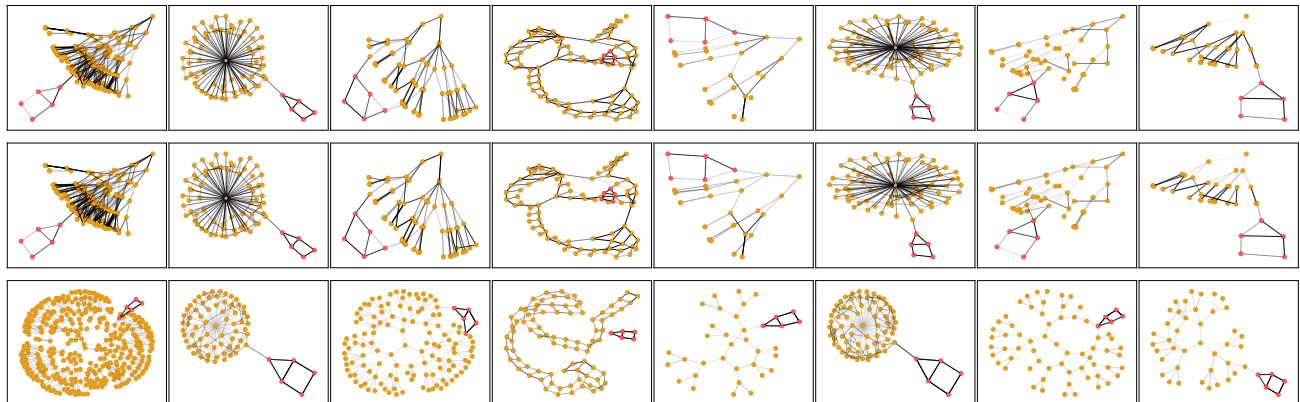

*Figure 10.* Learned interpretable subgraphs by GSAT (first row), GMT-LIN(second row) and TOPING(third row) on SPmotif0.9 class 1. Figures in the same column represent an identical graph. Nodes colored red are ground-truth explanations.

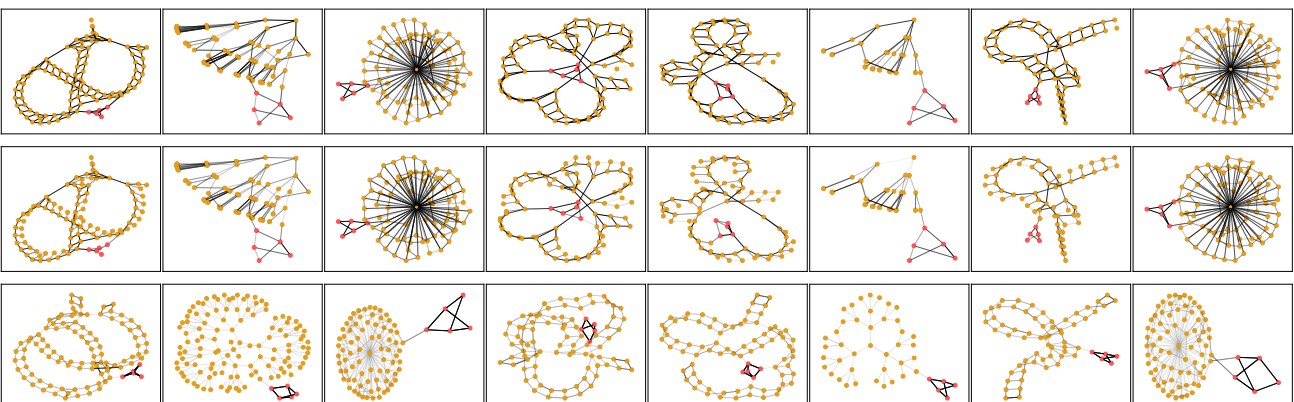

*Figure 11.* Learned interpretable subgraphs by GSAT (first row), GMT-LIN(second row) and TOPING(third row) on SPmotif0.9 class 2. Figures in the same column represent an identical graph. Nodes colored red are ground-truth explanations.

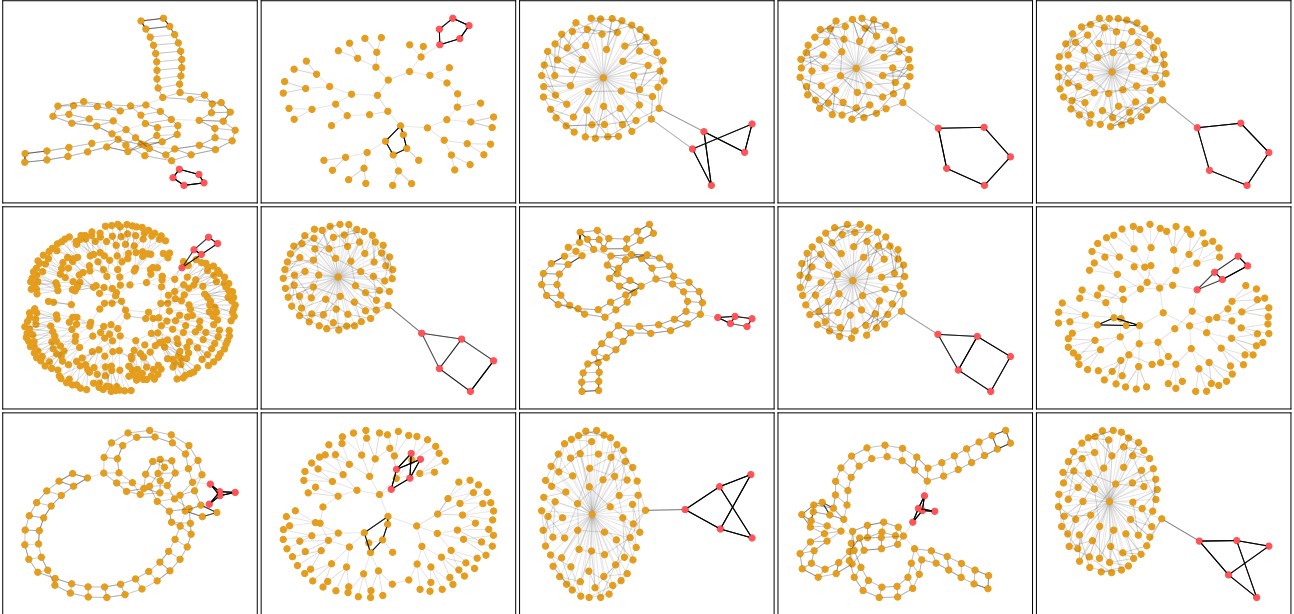

*Figure 12.* The rationals of SPmotif0.9 learned by TOPING. Figures in each row belong to the same category. Nodes colored red are ground-truth explanations.

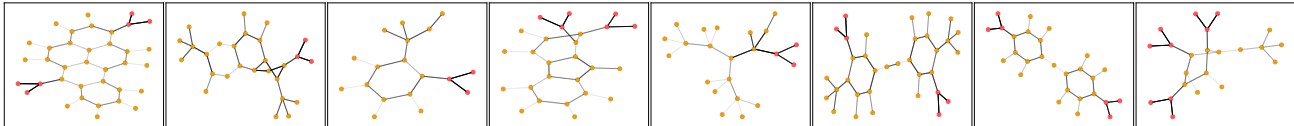

*Figure 13.* The rationals of Mutag learned by TOPING. Nodes colored red are ground-truth explanations.

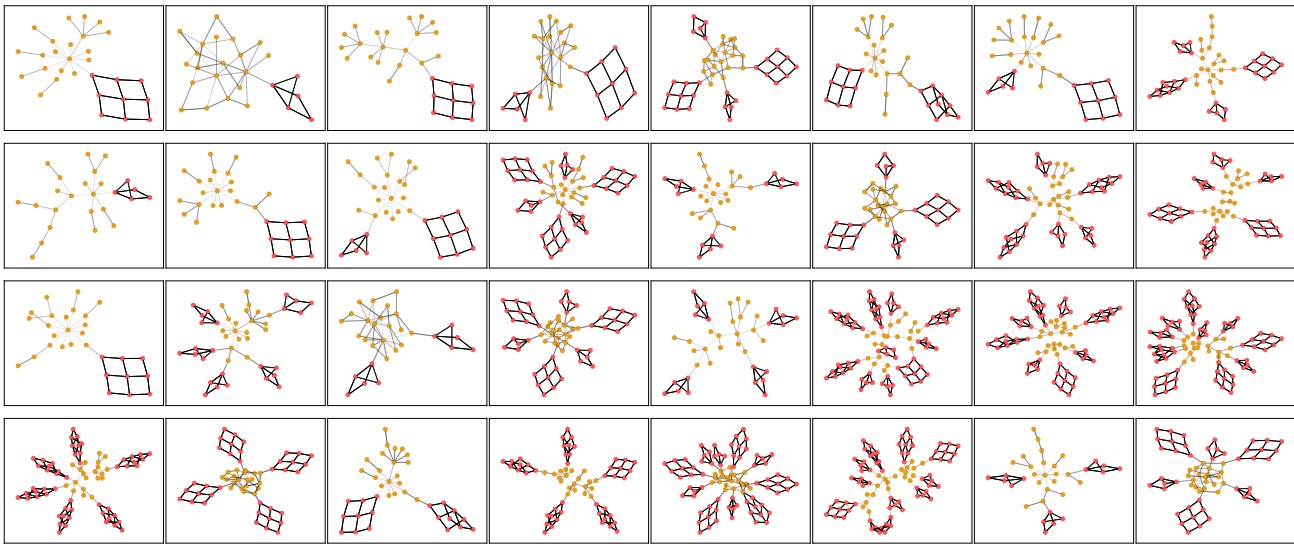

*Figure 14.* We trained TOPING on BA-HouseOrGrid-nRnd with $n = 4$ only, and test it on BA-HouseOrGrid-nRnd for $n = (2, 3, 5, 6)$. We still observe high prediction prediction(ACC=100%) and interpretation(AUC=100%) performance on test datasets. The figure illustrates interpretation results of BA-HouseOrGrid-nRnd for $n = (2, 3, 5, 6)$. Nodes colored red are ground-truth explanations.

