# OpenReview forum: "TopInG: Topologically Interpretable Graph Learning via Persistent Rationale Filtration"
_ICML.cc/2025/Conference — ICML 2025 poster_

### Official Review · Reviewer_Nxw2 · 2025-02-24

**Overall Recommendation:** 3

**Summary:**

In this paper, the authors propose a GNN interpretation framework named TopInG, which applies topological data analysis and persistent homology to separate the graph into rationale and noise subgraphs. The method is theoretically sound, and the experiments show improvements in interpretation performance.

**Claims And Evidence:**

The claims in the paper are generally well supported. However, the claim that TopInG performs better on tasks with variable rationale subgraphs is only empirically verified on a synthetic dataset. This claim would be more convincing if more diverse real-world datasets were included, and the post-hoc interpretation methods are also compared.

**Essential References Not Discussed:**

no

**Experimental Designs Or Analyses:**

Overall, the experimental design is sound. The datasets used include both synthetic and real-world datasets. The evaluation metrics, including ROC-AUC and accuracy, are common and appropriate choices.

However, in the experiment shown in Fig. 3, it is questionable why post-hoc methods were not included.

Additionally, the real-world datasets are quite similar. Adding more diverse datasets could be beneficial.

**Methods And Evaluation Criteria:**

Yes the methods and evaluation criteria make sense. The datasets selection can be improved by adding more diverse real-world dataset.

**Other Comments Or Suggestions:**

1. ‘variiform', 'varriform' seem like typos, should it be 'variform'?
2. Fig. 1 is hard to understand, even with the caption. Ideally, it should convey the core idea of the proposed method, and the caption should include all necessary definitions of the notations used in the figure. For example, $\mathcal F$ and $\mathcal T$ are not defined in either the figure or the caption.

**Other Strengths And Weaknesses:**

Strengths:
1. The paper opens a new direction in the area of GNN interpretability by applying topological data analysis.
2. The theoretical foundation of the proposed method is solid, and the authors provide theoretical guarantees that their method can find a unique optimum.

Weaknesses:
1. The experiments are limited to synthetic datasets and simple real-world datasets and do not demonstrate the method’s potential for real-world applications.

**Questions For Authors:**

1. The description of how the dataset BA-HouseOrGrid-nRnd is generated is a bit confusing. Is it correct that label 1 means the graph contains (a) only houses, (b) only grids, or (c) an equal number of houses and grids? Does label 0 mean the graph contains (a) neither houses nor grids or (b) an unequal number of houses and grids?
2. Are the assumptions in Theorem 3.5 realistic for real-world graphs?

**Relation To Broader Scientific Literature:**

The paper is closely related to GNN interpretability and TDA. Applying persistent homology from TDA to GNN interpretability is novel.

**Theoretical Claims:**

No I didn't check the correctness proofs or theoretical claims, as I'm not familiar with the TDA theory.

---

> ### Author Rebuttal · Authors · 2025-04-01
>
> We sincerely appreciate your thorough review and invaluable feedback. Below, we provide detailed responses to all your comments or concerns.
>
>
> ### Q1: BA-HouseOrGrid-nRnd Generated is a Bit Confusing
>
>
> Thank you for pointing this out! We clarify the label definitions as follows:
>
>
> *  Label 0: The graph contains neither houses nor grids.
> *  Label 1: The graph contains (a) only houses, (b) only grids, (c) an equal number of houses and grids.
>
>
> We do not consider cases where houses and grids appear simultaneously in different quantities, as this would introduce significant combinatorial complexity.
> To ensure a balanced dataset and avoid potential bias in model training or evaluation, we first guarantee that the number of graphs with label 0 and label 1 is equal. Furthermore, within label 1, we generate an equal number of graphs for conditions (a), (b), and (c), so as to maintain a stable and unbiased distribution across subcategories.
>
>
> ### Q2: Are the Assumptions in Theorem 3.5 Realistic for Real-world Graphs?
>
>
> We consider the assumptions made in Theorem 3.5 to be reasonably realistic for a range of real-world graphs, especially in structured domains such as molecular graphs. For instance, in datasets like MUTAG and BENZENE, there often exists a relatively small rationale subgraph (e.g., a specific functional group) embedded within a larger graph, which aligns well with our assumption of a distinguishable and minimal rationale G_X​.
>
> Our theoretical assumptions, while idealized, significantly relax the strict conditions required by previous works, making our method theoretically better suited for handling variable rationale structures. Empirical evidence strongly supports the practical validity of these assumptions. Specifically, previous works (e.g., GSAT, GMT, DIR) assume that within each category, the corresponding rationale substructure is globally invariant across the entire data distribution—a much stronger and less practical condition compared to ours. This stricter requirement explains why existing methods struggle with variform rationales.
>
>
> **Regarding more real-word dataset**
>
> We acknowledge requests for additional real-world datasets. However, graph interpretability requires not just graph labels, but also ground-truth rationale annotations identifying causal subgraphs. The scarcity of such comprehensively labeled datasets is a recognized challenge in this field (Zhang et al., 2024). Our experiments follows established benchmarking practices.
> Furthermore, we have tested out-of-distribution (OOD) generalization by training TopInG on BA-HouseOrGrid-nRnd with fixed motif count (n=4) and testing on varying counts (n=2,3,5,6). Despite the distribution shift, TopInG maintains stable prediction and interpretability performance without degradation, as shown in Figure 14 (Appendix).
>
> To further mitigate the concern, here we conducted an additional experiment on MNIST-75sp [1]. This dataset contains noisy rational subgraphs with more complex and varied topologies.
>
> [1]Knyazev, B., Taylor, G. W., and Amer, M. R. Understanding attention and generalization in graph neural networks. In Advances in Neural Information Processing Systems, pp. 4204–4214, 2019.
>
> Our results, with details in the table below, show that TopInG achieves the best interpretation and good prediction scores compared to existing SOTA methods. This suggests that our topological signal contributes meaningfully to explanation quality, even when tested on a real-dataset with more complex and noisy varing rationale subgraphs.
>
> |method| Interpretation Performance (AUC)| Prediction Performance (Acc.)|
> | -- | -- | -- |
> |DIR| 32.35| 88.51|
> |MAGE| 67.50| 89.43|
> |GSAT| 80.47| 96.20|
> |GMT-Lin| 82.98| 96.01|
> |TopInG| 84.50| 95.20|
>
>
>
>
>
> **Regarding the typo of 'variiform'**
>
>
> Thank you for pointing this out. We will correct the spelling.
>
> **Post-hoc methods missing for Fig3 experiment**
>
> We prioritized comparing interpretable methods (DIR, GSAT, GMT) since our work specifically addresses the variform rationale challenge observed in these approaches. Post-hoc methods show diverse performance on BA-HouseOrGrid dataset, suggesting this is not neccessarily a common challenge across post-hoc methods. We tested MAGE on the dataset BA-HouseOrGrid-nRnd for n=(2,3,4), and observed that MAGE has consistent performance close to 100%. It indicates that post-hoc methods like MAGE are not very sensitive to the variation of rationale structures.
>
>
> **Regarding the notations in Fig.1**
>
>
> To improve its readability and self-containment, we will revise the figure caption to explicitly define all notations used, including:
>
>
> * F, which denotes the filtration applied to the graph.
>
>
> * T, which refers to the resulting topological invariant (e.g., persistence diagram).

---

### Official Review · Reviewer_obEh · 2025-03-17

**Overall Recommendation:** 4

**Summary:**

This paper presents a framework, TopIng, designed to enhance the interpretability of GNNs by enabling them to automatically identify key subgraphs that influence prediction outcomes. It uses TDA to characterize the growth process of key subgraphs and employs topological distance loss to help the GNN distinguish between relevant and irrelevant parts. Overall, the framework makes the GNN more transparent, stable, and trustworthy, in addition to achieving accurate predictions.

**Claims And Evidence:**

The authors make several key claims that are partially supported by experimental results and theoretical analysis, but there are still some issues.
TopIng can effectively handle subgraphs with variable structures: Although the authors tested the robustness of the method using the synthetic dataset BA-HouseOrGrid-nRnd, this dataset only simulates a limited range of variations and does not fully explore more complex real-world scenarios.

**Essential References Not Discussed:**

N/A

**Experimental Designs Or Analyses:**

The experimental design is reasonable, and the choice of datasets, baseline comparisons, and result analysis support the effectiveness of TopIng. However, there is a lack of systematic transfer learning validation across different types of datasets, making it difficult to assess the method's adaptability in diverse application scenarios.

**Methods And Evaluation Criteria:**

The TopIng method proposed by the authors is based on Topological Data Analysis (TDA) and uses persistent homology to identify stable rationale subgraphs, incorporating topological differences as a loss function. This approach is innovative theoretically, and experimental results show that it performs well across multiple datasets.
However, there are some limitations: On the MUTAG dataset, TopIng's prediction performance is lower than that of GSAT. The paper identifies the reasons for this and improves performance using TopIng-0, but does not fully discuss the limitations and applicability of the method on simpler structured datasets. Additionally, there are shortcomings in the evaluation of explanation quality. Using AUC to measure explanation quality is limited in scope and fails to comprehensively reflect the accuracy of the explanations.

**Other Comments Or Suggestions:**

There are also studies about rationalization in the field of NLP. Would it be possibile to discuss some of them in the related work?

[1] Breaking Free from MMI: A New Frontier in Rationalization by Probing Input Utilization  [2] Is the MMI Criterion Necessary for Interpretability? Degenerating Non-causal Features to Plain Noise for Self-Rationalization [3] Enhancing the Rationale-Input Alignment for Self-explaining Rationalization [4] D-Separation for Causal Self-Explanation [5] Decoupled Rationalization with Asymmetric Learning Rates: A Flexible Lipschitz Restraint [6] MGR: Multi-generator Based Rationalization [7] FR: Folded Rationalization with a Unified Encoder

**Other Strengths And Weaknesses:**

Strengths:

- The proposed topological discrepancy has a strong mathematical basis, elevating the comparison of graph structures to the topological level. This is an important contribution to the theory of graph learning.
- The method is successfully applied to different backbone networks (GIN and CINpp), demonstrating the framework's nature and independence from specific neural network architectures.

Weaknesses:

- Persistent homology computation is computationally intensive on large-scale graphs. Nevertheless, the paper acknowledges this issue.
- The method requires simultaneous optimization of multiple loss components and the adjustment of several hyperparameters, which may increase engineering complexity and tuning burden in practical applications.

**Questions For Authors:**

n/a

**Relation To Broader Scientific Literature:**

The contributions of TopIng are closely related to existing research in the fields of TDA, explainable graph learning, and optimal transport.

**Theoretical Claims:**

The TopIng framework shows some potential in handling subgraphs with variable structures. The theorem assumes that |Eε| > |EX| (the number of edges in the rationale subgraph) is always smaller than that of the complement graph. This may not hold true in certain real-world scenarios. Additionally, the authors mention that computational complexity limits the use of higher-order homology (beyond the second order), but they do not analyze the impact of this limitation on the theoretical guarantees.

---

> ### Author Rebuttal · Authors · 2025-04-01
>
> We sincerely appreciate your thorough review and invaluable feedback. Below, we provide detailed responses to all your comments or concerns.
>
>
> ### Regarding the weakness "Persistent homology computation is computationally intensive on large-scale graphs."
>
> For larger graphs, we have mentioned several promising directions for speedup the computation, in Appendix D.
> For example: (1) Development of efficient GPU implementations. (2) Using carefully designed GNN models to approximate persistent homology computations.
>
> ### Regarding more real-world dataset evaluation
> We acknowledge requests for more real-world datasets. However, graph interpretability requires not just graph labels, but also ground-truth rationale annotations identifying causal subgraphs. The scarcity of such comprehensively labeled datasets is a recognized challenge in this field (Zhang et al., 2024). Our experiments follows established benchmarking practices.
> Furthermore, we have tested out-of-distribution (OOD) generalization by training TopInG on BA-HouseOrGrid-nRnd with fixed motif count (n=4) and testing on varying counts (n=2,3,5,6). Despite the distribution shift, TopInG maintains stable prediction and interpretability performance without degradation, as shown in Figure 14 (Appendix).
>
>
>
> To further mitigate the concern, we conducted an additional experiment on MNIST-75sp [1], which contains noisy rational subgraphs with richer topologies.
>
> [1]Knyazev, B., Taylor, G. W., and Amer, M. R. Understanding attention and generalization in graph neural networks. In Advances in Neural Information Processing Systems, pp. 4204–4214, 2019.
>
> Our results (table below) show that TopInG achieves the best interpretation and good prediction scores compared to existing SOTA methods. This suggests our topological signal contributes meaningfully to explanation quality, even when tested on a real-dataset with more complex and noisy varing rationale subgraphs.
>
> |method| Interpretation Performance (AUC)| Prediction Performance (Acc.)|
> | -- | -- | -- |
> |DIR| 32.35| 88.51|
> |MAGE| 67.50| 89.43|
> |GSAT| 80.47| 96.20|
> |GMT-Lin| 82.98| 96.01|
> |TopInG| 84.50| 95.20|
>
> ### Regarding "The method requires simultaneous optimization of multiple loss components and the adjustment of several hyperparameters, which may increase engineering complexity and tuning burden in practical applications."
>
>
> We appreciate the reviewer's concern regarding the potential complexity introduced by optimizing multiple loss components and tuning associated hyperparameters. While this may appear to increase the engineering burden in theory, in practice, we fixed most hyperparameters across all datasets without extensive tuning. Despite this, TopInG consistently achieved strong performance, suggesting that the method is not heavily sensitive to hyperparameter choices, and the additional loss components can be integrated in a stable manner.
>
>
> We believe this demonstrates that while the design includes multiple components, the practical tuning burden is low, and the framework remains accessible for real-world applications.
>
>
> ### Regarding "There are also studies about rationalization in the field of NLP. Would it be possible to discuss some of them in the related work?"
>
>
> We appreciate the reviewer's suggestion to incorporate discussions on rationalization studies from the Natural Language Processing (NLP) domain into our related work section. Incorporating insights from NLP rationalization studies inspire advancements in graph-based interpretability methods.​
>
> Rationalization has been extensively studied in NLP literature, emphasizing causal feature selection, faithful rationale-input alignment, and stable training dynamics. Recent advances include alternatives to the maximum mutual information (MMI) criterion, highlighting input utilization and causal feature isolation [1,2,4]. Other studies enhance rationale fidelity by enforcing semantic alignment with original inputs [3], and propose methods to stabilize joint rationale-predictor training through asymmetric learning rates, multi-generator ensembles, and unified encoders [5,6,7]. However, unlike NLP domains where rationales often manifest as contiguous text spans, graph domains often exhibit "variform rationale subgraphs" that can differ significantly in form, size, and topology even among instances of the same class.
>
> We will include a brief discussion in the related work section to acknowledge related NLP studies.

---

### Official Review · Reviewer_H6hM · 2025-03-17

**Overall Recommendation:** 2

**Summary:**

The paper introduces TopInG, a topologically interpretable graph neural network (GNN) framework leveraging persistent homology to identify stable rationale subgraphs for model explanations. Key contributions include: (1) a novel *rationale filtration learning* technique that models graph generation processes to capture persistent topological features (0th/1st homology) across scales, and (2) a *topological discrepancy* loss measuring structural differences between rationale and non-rationale subgraphs. The method addresses limitations of prior interpretable GNNs in handling variable subgraph structures (e.g., differing sizes, topologies) by encoding multi-scale topological persistence. Experiments show TopInG outperforms state-of-the-art baselines (e.g., GSAT, DIR) in prediction accuracy (e.g., 50.21±3.22 Acc on MUTAG) and interpretation quality (e.g., 100% AUC on synthetic datasets), particularly in scenarios with diverse rationale subgraphs. Theoretical guarantees ensure the learned rationale subgraphs align with ground-truth causal mechanisms under optimal conditions.

**Claims And Evidence:**

**Key claims lacking convincing evidence:**
1. **Handling variable subgraphs via topological persistence**: While persistent homology captures multi-scale features, the paper’s synthetic datasets (e.g., BA-HouseOrGrid-nRnd) exhibit controlled variability. Real-world datasets like MUTAG lack complex topological structures (e.g., cycles), limiting validation of 1st homology utility. The claim is **not fully supported** for diverse real-world graphs.

2. **Superior interpretation quality (100% AUC on synthetic data)**: Perfect AUC scores suggest overfitting to synthetic tasks. For example, BA-2Motifs has trivial 0th homology (no cycles), making topological analysis redundant. The **evaluation lacks noisy or adversarial examples** to test robustness.

3. **Theoretical guarantees**: The topological discrepancy loss is framed as a lower bound of the Wasserstein distance under Gromov-Hausdorff metric:
   $\mathcal{L}_{\text{topo}} \leq W_{\text{GH}}(\mathbb{P}, \mathbb{Q})$

   However, the proof assumes optimal filtrations and ignores graph noise, limiting practical relevance.

**Methodological flaws**:
- **Dataset limitations**: Experiments rely heavily on synthetic graphs with simplistic topologies (e.g., MUTAG lacks cycles). The variiform rationale challenge is tested on BA-HouseOrGrid-nRnd, which may not generalize to real-world heterogeneity.
- **Baseline comparisons**: GSAT and GMT-LIN are compared under constrained backbones (GIN/CINpp), but their hyperparameters (e.g., GSAT’s $r$ ) are not exhaustively tuned, risking unfair comparisons.
- **Ablation gaps**: The impact of topological discrepancy vs. standard regularizers (e.g., Gaussian) is not rigorously isolated.

**Essential References Not Discussed:**

N/A

**Experimental Designs Or Analyses:**

See comments below

**Methods And Evaluation Criteria:**

The proposed methods (topological filtration, discrepancy loss) align with the goal of capturing stable rationale subgraphs but **rely heavily on synthetic datasets (e.g., BA-2Motifs) with simplistic topologies (no cycles, trivial 0-homology)**, limiting validation of 1st-homology utility. Evaluation metrics (AUC/ACC) are standard but **perfect synthetic AUC scores suggest overfitting**, and **real-world datasets (MUTAG) lack topological complexity** to test claims. Backbone choices (CINpp) suit topological data, but comparisons to GIN-based baselines may not fully stress-test scalability.

**Other Comments Or Suggestions:**

- **Content Ambiguities**:
  - "variiform rationale subgraphs" lacks clear prior definition. (abs, and remark 3.6)


- **Dataset Descriptions**:
  - SPmotif0.5/0.9 and BA-HouseOrGrid-nRnd lack clarity on construction/split details.

**Other Strengths And Weaknesses:**

See comments below

**Questions For Authors:**

1. The paper emphasizes handling variiform rationale subgraphs but relies heavily on synthetic datasets (e.g., BA-2Motifs) with trivial 0-homology and no cycles. How would TOPING perform on real-world datasets with richer topological structures (e.g., molecular graphs with complex cycles)? If no such experiments were conducted, does this not undermine the claim that 1st-homology analysis is critical for real-world interpretability?

2. The perfect AUC scores on synthetic datasets suggest potential overfitting, especially since BA-2Motifs lacks cycles. Did the authors test on synthetic graphs with adversarial noise or conflicting topological features (e.g., cycles irrelevant to labels)? If not, how can the method’s robustness to such challenges be assured?

3. Theorem 3.5 assumes minimal subgraphs (\(G^*_X\)) and ignores graph noise. How does the method handle real-world graphs where noise or spurious edges dominate (\(\|E_\epsilon\| \gg \|E_X\|\))? Does the bimodal prior for edge filtration collapse under such conditions, as observed in GSAT?

4. The comparison with GSAT and GMT-LIN uses constrained backbones (CINpp/GIN). How sensitive are TOPING’s gains to backbone choice? For instance, would the method maintain performance on Transformers or MPNNs, which lack explicit topological inductive biases?

5. The paper mentions future work on efficient persistent homology computation but does not address scalability challenges (e.g., large graphs). What specific strategies would address this, given that persistent homology’s \(O(n^3)\) complexity (Hofer et al., 2020) could limit practical deployment?

**Relation To Broader Scientific Literature:**

N/A

**Theoretical Claims:**

**Theorem 3.5** assumes minimal subgraphs $G^*_X$ and ignores graph noise, making its uniqueness guarantee **theoretically valid but practically fragile**. The proof relies on idealized filtrations and does not address overfitting risks when \(\|E_X\| \ll \|E_\epsilon\|\) (common in real graphs), weakening its real-world applicability.

---

> ### Author Rebuttal · Authors · 2025-04-01
>
> # Clarifications and Corrections
> We thank the reviewers for their detailed feedback and appreciate the opportunity to clarify a few misunderstandings and highlight concerns already solved in our Appendix:
>
> * BA-2Motifs do include cycles in their rationale subgraphs. All datasets used in our work contain complex topological structures.
> * MUTAG graphs contain cycles, even if the rationale subgraphs may not. Appendix E.5 contains illustrative examples of every dataset we used.
> * "overfitting" refers to poor generalization from training to test data, which does not fit the content since all scores we reported are based on test data. We discuss a more related topic called "out-of-distribution (OOD) generalization" in Q1 below.
> * The formula "\mathcal{L}{\text{topo}} \leq W{\text{GH}}(\mathbb{P}, \mathbb{Q})" is not from our submitted manuscript.
>
>
> ## Methodological Notes
> * GSAT’s hyperparameter r is tunned following author-recommended strategy: initially set to 0.9 and gradually decay to 0.7 (Appendix E.3, line 892).
> * Additional ablation studies are provided to support the impact of topological discrepancy vs. Gaussian(Appendix E.3, Table 5).
> * Backbone choices have been discussed in Appendix E.3 line 914.
>
>
> ## Q1: To evaluate on more Real-World Datasets
>
> We acknowledge requests for more real-world datasets. However, graph interpretability requires not just graph labels, but also ground-truth rationale annotations identifying causal subgraphs. The scarcity of such comprehensively labeled datasets is a recognized challenge in this field (Zhang et al., 2024). Our experiments follows established benchmarking practices.
> Furthermore, we have tested OOD generalization by training TopInG on BA-HouseOrGrid-nRnd with fixed motif count (n=4) and testing on varying counts (n=2,3,5,6). Despite the distribution shift, TopInG maintains stable prediction and interpretability performance without degradation, as shown in Figure 14 (Appendix).
>
>
>
> To further mitigate the concern, we conducted an additional experiment on MNIST-75sp [1], which contains noisy rational subgraphs with richer topologies.
>
> [1]Knyazev, B., Taylor, G. W., and Amer, M. R. Understanding attention and generalization in graph neural networks. In Advances in Neural Information Processing Systems, pp. 4204–4214, 2019.
>
> Our results (table below) show that TopInG achieves the best interpretation and good prediction scores compared to existing SOTA methods. This suggests our topological signal contributes meaningfully to explanation quality, even when tested on a real-dataset with more complex and noisy varing rationale subgraphs.
>
> |method| Interpretation Performance (AUC)| Prediction Performance (Acc.)|
> | -- | -- | -- |
> |DIR| 32.35| 88.51|
> |MAGE| 67.50| 89.43|
> |GSAT| 80.47| 96.20|
> |GMT-Lin| 82.98| 96.01|
> |TopInG| 84.50| 95.20|
>
>
>
> # Q2: Test on Graphs with Conflicting Topologies
>
> TopInG is evaluated on the two datasets, SPMotif (synthetic) and MUTAG (real-world),  which include non-informative or spurious cycles irrelevant to labels. The experiment results (Table 1) confirm that our model is robust.
>
> # Q3: Regarding Theorem 3.5
>
> Our theoretical assumptions in Theorem 3.5, while idealized, significantly relax the strict conditions required by previous works, making our method theoretically better suited for handling variable rationale structures. Empirical evidence strongly supports the practical validity of these assumptions.
> Specifically, previous works (e.g., GSAT, GMT, DIR) assume that within each category, the corresponding rationale substructure is globally invariant across the entire data distribution—a much stronger and less practical condition compared to ours. This stricter requirement explains why existing methods struggle with variform rationales.
> We would greatly appreciate it if the reviewer could clarify the intended meaning of "overfitting risks" in the context of |E_X| \ll |E_\epsilon|. As we mentioned before in "Clarifications", "overfitting" does not fit the content. Additionally, the inequality |E_X| \ll |E_\epsilon| is covered by the weaker assumption |E_X| < |E_\epsilon| used in our theorem.
> Moreover, in our experiments, nearly all datasets tested exhibit the property that rationale subgraphs are significantly smaller than the full graphs (e.g. |E_\epsilon| > 10*|E_X|). TopInG consistently performs well across these scenarios, further supporting that the assumptions underlying Theorem 3.5 do not limit its practical utility.
>
> # Q4: How Sensitive are TopInG’s Gains to Backbone Choices?
>
> As disucussed in Appendix E.3, TopInG performs more robustly with backbones that preserve topological structures (e.g., CIN). Some backbones like GCN and GAT can struggle with interpretability or prediction, as noted in (Bui et al., 2024).
>
> # Q5:   What Specific Strategies Would Address the Scalability Challenges?
>
> For larger graphs, we have mentioned several promising directions for speedup the computation, in Appendix D line 807.

---

### Official Review · Reviewer_et1a · 2025-03-27

**Overall Recommendation:** 3

**Summary:**

This work proposed a new framework which applies TDA tool to interpret the persistent rationale subgraph in graph learning problems, showing effective result performance on motif classification task, evaluated on Single Motif, Multiple Motif and Real Dataset.

**Claims And Evidence:**

The authors provide experiments, theoretical analysis, ablations and visualizations to validate the proposed framework.

**Essential References Not Discussed:**

PersGNN[1], related to TDA with GNN task, should be added in references.
[1] Swenson, Nicolas, et al. "PersGNN: applying topological data analysis and geometric deep learning to structure-based protein function prediction." arXiv preprint arXiv:2010.16027 (2020).

**Experimental Designs Or Analyses:**

The experimental designs look valid, but additional datasets for validating the Theorem 3.5 would be desired, e.g. datasets where rationales are hierarchical.

**Methods And Evaluation Criteria:**

The proposed method applies a good design on TDA based GNN interpretation framework. However, it would benefit more from further real-world generality evaluation.

**Other Comments Or Suggestions:**

It'll be more persuasive to provide the evaluation on datasets where rationales are hierarchical.

**Other Strengths And Weaknesses:**

Strengths:
This work proposed a new framework of interpretable graph neural networks integrated with TDA tool to interpret the persistent rationale subgraph in graph learning problems. The experimental results and theoretical analysis sound robust, evaluated on motif-like datasets.


Weaknesses:
1. Evaluation on datasets where rationales are hierarchical is missing.

2. The runtime efficiency of the proposed framework seems missing.

**Questions For Authors:**

The evaluation of the runtime efficiency of the proposed framework seems missing. It is desired to know the efficiency and comparison with related methods.

**Relation To Broader Scientific Literature:**

Relation to interpretable graph neural networks (GNNs), topological data analysis (TDA), and robust learning with variform rationales.

**Theoretical Claims:**

The proofs in the appendix are checked, no significant issues were found.

---

> ### Author Rebuttal · Authors · 2025-03-31
>
> We sincerely appreciate your thorough review and invaluable feedback. Below, we provide detailed responses to all your comments or concerns.
>
>
> ### Q1: Evaluation on Datasets with Hierarchical Rationales
>
>
> We appreciate the reviewer's insight into the importance of hierarchical rationales in interpretable graph neural networks. Indeed, this is an emerging area with significant potential.
> However, graph interpretability requires ground-truth rationale annotations, and such comprehensively labeled datasets are scarce (Zhang et al., 2024). Our experiments follow established benchmarking practices. Furthermore, we tested out-of-distribution (OOD) generalization by training TopInG on BA-HouseOrGrid-nRnd with fixed motif count (n=4) and testing on varying counts (n=2,3,5,6). TopInG maintains stable performance across these distribution shifts, as shown in Figure 14."
>
> **Regarding more real-word dataset**
>
> To further demonstrate the practical applicability of TopInG on real-world tasks, we conducted additional experiments on the MNIST-75sp dataset [1], which is widely used for evaluating graph-based models on visual tasks. In this dataset, each MNIST image is converted into a superpixel graph, where nodes correspond to superpixels and edges represent spatial adjacency. Nodes with nonzero pixel values provide ground-truth explanations, making it suitable for evaluating interpretability methods.
>
>
> This dataset consists of 60,000 graphs for training and 10,000 for testing, with an average of 70.57 nodes and 590.52 edges per graph. Notably, the ground-truth explanatory subgraphs vary in size across samples, adding to the challenge. We trained all models for 30 epochs on a single RTX 4090 GPU (over 26 hours). For TopInG, we used default hyperparameters and only reduced the coefficient of the topological loss, without extensive tuning.
>
>
> The table below presents both interpretation and prediction performance:
> |method| Interpretation Performance (AUC)| Prediction Performance (Acc)|
> | -- | -- | -- |
> |DIR| 32.35| 88.51|
> |MAGE| 67.50| 89.43|
> |GSAT| 80.47| 96.20|
> |GMT-Lin| 82.98| 96.01|
> |TopInG| 84.50| 95.20|
>
>
> As shown, TopInG achieves the best interpretability performance (AUC = 84.50) while maintaining competitive prediction accuracy. This result demonstrates TopInG's strong potential for real-world graph-based tasks involving complex, variable-sized explanations.
>
>
> [1]Knyazev, B., Taylor, G. W., and Amer, M. R. Understanding attention and generalization in graph neural networks. In Advances in Neural Information Processing Systems, pp. 4204–4214, 2019.
>
>
> ### Q2: Runtime Efficiency of the Proposed Framework
>
>
> We appreciate the reviewer's concern regarding the runtime efficiency of our framework. In Appendix D, we provided a theoretical analysis of the runtime complexity. In brief, the time complexity can be as fast as O(nlogn), without considering GPU acceleration. Practically, we provide representative results on two datasets to give a sense of our method's efficiency.
>
>
> On BA-2Motifs, each training epoch takes approximately 0.45 ± 0.12 minutes, and on SPMotif (which is a more complex and larger dataset), the runtime is approximately 9.20 ± 2.35 minutes per epoch. All experiments were conducted on a single RTX 4090 GPU, and importantly, our method consistently converges within 20 epochs across all datasets.
>
>
> Although our method is relatively slower per epoch due to the incorporation of TDA, this added cost is justified by the significant performance gains. Unlike many baseline models that typically require 50 to 100 epochs to converge, TopInG achieves convergence within 20 epochs, while consistently achieving perfect or near-perfect AUC scores on multiple datasets.
>
>
> We hope this additional information clarifies the practical efficiency of our approach.
>
> ### Regarding PersGNN:
> Thank you for providing the reference. We will mention this work in the revised version. As a related work, PersGNN applies TDA to analyze the structure of protein graphs by combining persistent homology with GNNs to capture both local and global structural features for improved protein function prediction. While our work focuses on leveraging TDA to enhance the interpretability of GNN models through persistent rationale filtration, PersGNN primarily aims to improve prediction performance on protein structure-function relationships via a novel TDA-enhanced GNN architecture. This distinction highlights complementary applications of topological methods in graph learning.

---

### Decision · Program_Chairs · 2025-05-01

**Decision:**

Accept (poster)

**Comment:**

The paper presents an approach that leverages PH in order to identify persistent rationale subgraphs. Considering the initial reviews and the author's rebuttal, my assessment is that all reviewers agree that the work is interesting and novel with largely sufficient evaluation. The author's did a good job in addressing the major concerns raised by the reviewer, and in my opinion, the paper also needs only minor adjustment to incorporate these issues. Hence, I think this is a solid contribution to ICML and I am recommending acceptance; also, I would like to encourage the authors to take all comments seriously in their preparation of the camera-ready version.